# The selectivity of the Na$^+$/K$^+$-pump is controlled by binding site protonation and self-correcting occlusion

**Huan Rui[1], Pablo Artigas[2], Benoît Roux[1]\***

[1]Department of Biochemistry and Molecular Biology, The University of Chicago, Chicago, United States; [2]Department of Cell Physiology and Molecular Biophysics, Texas Tech University Health Sciences Center, Lubbock, United States

**Abstract** The Na$^+$/K$^+$-pump maintains the physiological K$^+$ and Na$^+$ electrochemical gradients across the cell membrane. It operates via an 'alternating-access' mechanism, making iterative transitions between inward-facing (E$_1$) and outward-facing (E$_2$) conformations. Although the general features of the transport cycle are known, the detailed physicochemical factors governing the binding site selectivity remain mysterious. Free energy molecular dynamics simulations show that the ion binding sites switch their binding specificity in E$_1$ and E$_2$. This is accompanied by small structural arrangements and changes in protonation states of the coordinating residues. Additional computations on structural models of the intermediate states along the conformational transition pathway reveal that the free energy barrier toward the occlusion step is considerably increased when the wrong type of ion is loaded into the binding pocket, prohibiting the pump cycle from proceeding forward. This self-correcting mechanism strengthens the overall transport selectivity and protects the stoichiometry of the pump cycle.

**\*For correspondence:** roux@uchicago.edu

**Competing interests:** The authors declare that no competing interests exist.

## Introduction

The Na$^+$/K$^+$-pump is a primary active membrane transporter present in nearly all animal cells. It belongs to the P-type ATPase family, which utilizes the energy released from ATP hydrolysis to move ions against their concentration gradients across a membrane barrier. The ion species transported by the pump under physiological conditions are Na$^+$ and K$^+$. Like many other membrane transporters, the Na$^+$/K$^+$-pump works according to an 'alternating-access' ion transport mechanism, with the bound ions accessible from only one side of the membrane at a time. The consensus scheme of the pump cycle is known as the 'Albers-Post' cycle (*Albers, 1967*; *Post et al., 1969*). It involves two major conformations, E$_1$ and E$_2$, with inward- and outward-facing ion binding sites, respectively. In each cycle, the E$_1$ conformation binds three Na$^+$ from the cytosol and exports them using the energy from ATP hydrolysis. After external release of Na$^+$, binding of extracellular K$^+$ promotes the structural transition to the occluded E$_2$ state, which finally imports two K$^+$ as binding of ATP returns the pump to the E$_1$ conformation. E$_1$/E$_2$ conformational change and the accompanied electrogenic active transport are facilitated by the phosphorylation and dephosphorylation of an aspartate residue in the cytoplasmic domain, which is a hallmark of the P-type ATPase family (*Axelsen et al., 1998*).

The presence of the Na$^+$/K$^+$-pump is essential for excitability and secondary active transport. More than 40% of the energy produced in mammals is consumed by the Na$^+$/K$^+$-pump (*Milligan et al., 1985*). Although it is a machine designed for such a precise and important function, it has been shown that many cations, including congeners of K$^+$ and some organic cations, can bind to the same sites used by the pump to bind and transport K$^+$ ions (*Mahmmoud et al., 2015*;

**eLife digest** A protein called the sodium-potassium pump resides in the membrane that surrounds living cells. The role of this protein is to 'pump' sodium and potassium ions across the membrane to help restore their concentration inside and outside of the cell. About 25% of the body's energy is used to keep the pump going, rising to nearly 70% in the brain. Problems that affect the pump have been linked to several disorders, including heart, kidney and metabolic diseases, as well as severe neurological conditions.

The sodium-potassium pump must be able to effectively pick out the correct ions to transport from a mixture of many different types of ions. However, it was not clear how the pump succeeds in doing this efficiently. Rui et al. have now used a computational method called molecular dynamics simulations to model how the sodium-potassium pump transports the desired ions across the cell membrane.

The pump works via a so-called 'alternating-access' mechanism, repeatedly transitioning between inward-facing and outward-facing conformations. In each cycle, it binds three sodium ions from the cell's interior and exports them to the outside. Then, the pump binds to two potassium ions from outside the cell and imports them inside. Although the bound sodium and potassium ions interact with similar binding sites in the pump, the pump sometimes preferentially binds sodium, and sometimes potassium. The study performed by Rui et al. shows that this preference is driven by how protons (hydrogen ions) bind to the amino acids that make up the binding site.

The simulations also suggest that the pump uses a 'self-correcting' mechanism to prevent the pump from transporting the wrong types of ions. When incorrect ions are present at the binding sites, the pump cycle pauses temporarily until the ions detach from the pump. Only when the correct ions are bound will the pump cycle continue again.

In the future, Rui et al. hope to use long time-scale molecular dynamics simulations to show the conformational transition in action. In addition, the 'self-correcting' mechanism will be directly tested by letting the wrong and correct ions compete for the binding sites to see whether the pump will transport only the correct ions.

*Ratheal et al., 2010*). Competitive binding between $Na^+$ and other cations at the cytoplasmic side of the membrane has also been observed (*Schneeberger et al., 2001*). An unsolved puzzle, therefore, is how the $Na^+$/$K^+$-pump is able to recognize and accept two $K^+$ from the extracellular matrix, where $Na^+$ concentration is much higher, and how it selectively binds $Na^+$ from the cytoplasm to keep the pump cycle running forward.

Structural studies have provided tremendous insights into the function of the $Na^+$/$K^+$-pump, which consists of two obligatory subunits, α (catalytic) and β (auxiliary), and sometimes a tissue specific regulatory subunit from the FXYD protein family (*Geering, 2006*; *Mercer et al., 1993*). The transmembrane region of the α-subunit contains the ion binding sites within its 10 helices (called M1-M10). Recent crystal structures of the pump in its $E_1$ and $E_2$ states reveal the locations of the three $Na^+$ binding sites in $E_1$ and the two $K^+$ binding sites in $E_2$ (*Kanai et al., 2013*; *Laursen et al., 2013*; *Morth et al., 2007*; *Shinoda et al., 2009*). From structural alignments based on the heavy atom positions in the binding sites, it becomes clear that the binding pocket harboring sites I and II in $E_1$ overlaps with those in $E_2$ (*Figure 1*). Site III is only formed in $E_1$. It is located between the transmembrane helices M5, M6, and M8 and is thought to exclusively bind $Na^+$ but appears to catalyze $H^+$ transport (*Poulsen et al., 2010*; *Ratheal et al., 2010*) in a manner that presents a complex dependence on the concentrations of $Na^+$, $K^+$, and $H^+$ (*Mitchell et al., 2014*). Previous studies have indicated that protonation at the ion binding pocket may play a role in the selectivity of these sites for $K^+$ when the pump is in its $E_2$ state (*Yu et al., 2011*). Biochemical assays on the mutant D926N, which is often used as a surrogate for protonated D926, also show that it induces distinct effects on $Na^+$ and $K^+$ binding (*Einholm et al., 2010*; *Jewell-Motz et al., 1993*), suggesting a potential change in its protonation state upon the $E_1$/$E_2$ transition. Taken together, the evidence, although indirect, suggests the possibility of an $E_1$ specific protonation state that favors $Na^+$ binding to the pump. The nature of this protonation state is, however, unclear.

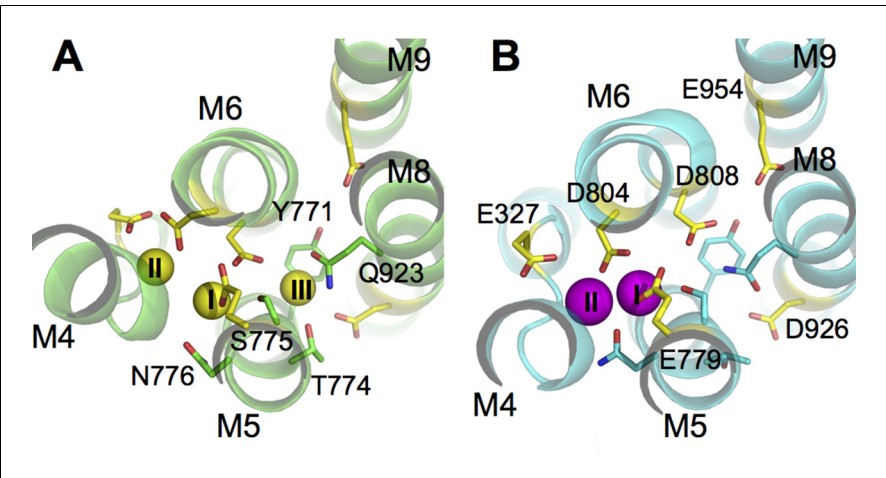

**Figure 1.** The ion binding sites in (**A**) $Na_3 \cdot E_1 \cdot (ADP \cdot Pi)$ (PDBID 3WGV) and (**B**) $E_2(K_2)$ (PDBID 2ZXE) states. Only the transmembrane helices M4, M5, M6, M8, and M9 from the α subunit are shown. Residues in the binding site are highlighted in stick presentation with those protonatable colored in yellow. $Na^+$ (yellow) and $K^+$ (magenta) ions are in spheres. The binding site number indices are presented on top of the ions. The view is from the extracellular side towards the intracellular side. The figure is produced with PyMOL (**DeLano, 2002**).

In the present study, we started with the recently published crystal structure of the $Na^+/K^+$-pump in a partially occluded $Na_3 \cdot E_1(ADP \cdot Pi)$ state and used molecular dynamics (MD) simulations to show that there is a correlation between the binding pocket protonation and the $Na^+$ selectivity in $E_1$. The binding sites in $Na_3 \cdot E_1(ADP \cdot Pi)$ were tested one by one by 'alchemically transforming' the bound $Na^+$ into a $K^+$ while keeping the other two sites occupied by $Na^+$. A similar approach was previously used to study the selectivity in other conformational states of the pump (**Yu et al., 2011**) (see also Materials and methods section). The results show that the $Na^+$ selectivity at all three sites is realized only when a specific set of binding pocket residues are protonated. This set of residues is, by comparison, different from that in the $K^+$ selective $E_2$ state. The implication is that the protonation state and the selectivity of the pump are tightly coupled; when the pump undergoes a transition between $E_1$ and $E_2$, a protonation state switch occurs. The present findings also show that the effective selectivity of the pump is reinforced by a self-correcting mechanism, which prevents the occlusion step on either side of the membrane if the wrong type of ions were loaded into the binding sites. This mechanism ensures that counterproductive transport cycles do not occur.

## Results

### pKa of the binding pocket protonatable residues

The crystal structures of the pump in its $E_1$ (PDBID 3WGV [**Kanai et al., 2013**]) and $E_2$ (PDBID 2ZXE [**Shinoda et al., 2009**]) conformations show that there is a large structural overlap between the $Na^+$ and $K^+$ binding pockets. The sites I and II are in the main binding chamber formed by helices M4, M5, and M6, while site III, located in between M5, M6, and M8 (**Figure 1**), is $Na^+$ exclusive and appears in $E_1$ only. The coordination of $Na^+$ and $K^+$ at sites I and II are similar. Many residues are found to coordinate both $Na^+$ and $K^+$ at these two sites in $E_1$ and $E_2$ (**Table 1**). Although the structural difference between the ion binding pockets in $E_1$ and $E_2$ may not be particularly large, the local physicochemical environment can display considerable variations. In particular, the latter could affect the pKa and the protonation states of the six protonatable residues in the binding pocket (**Figure 1**). Their pKa values calculated using PROPKA3.1 (**Olsson et al., 2011**) are listed in **Table 2**.

It is important to note that the pKa values calculated with empirical methods like PROPKA are sensitive to local structural perturbations. Even when the structures assume the identical conformational state and are taken from the same asymmetric unit from a crystal, the pKa values of the same residue can differ by more than one pH unit. For example, the pKa of E779 is 9.9 in chain A of the

**Table 1.** Atoms coordinating the binding site ions in the crystal structures and from the MD simulations. O is the backbone carbonyl oxygen atom. OG and OG1 are the hydroxyl oxygen atoms in serine and threonine. OD1 and OD2 are the carboxyl oxygen atoms in asparate. OE1 and OE2 are the carboxyl oxygen atoms in glutamate. OH2 is the water oxygen.

| | Site I | | | | Site II | | | | Site III | | | |
|---|---|---|---|---|---|---|---|---|---|---|---|---|
| $E_1$ | x-ray | | MD | | x-ray | | MD | | x-ray | | MD | |
| | A323 | O | T772 | OG1 | V322 | O | E779 | OE1 | Y771 | O | Y771 | O |
| | E779 | OE1 | T772 | O | V325 | O | D804 | OD1 | T774 | O | T774 | O |
| | D808 | OD1 | N776 | OD1 | E327 | OE2 | D808 | OD1 | Q923 | OE1 | Q923 | OE1 |
| | | | D808 | OD1 | D804 | OD1 | Water | OH2 | | | D926 | OD1 |
| | | | D808 | OD2 | Water | OH2 | | | | | | |
| | Site I | | | | Site II | | | | | | | |
| $E_2$ | x-ray | | MD | | x-ray | | MD | | | | | |
| | T772 | O | S775 | OG | V322 | O | A323 | O | | | | |
| | S775 | OG | N776 | OD1 | V325 | O | V325 | O | | | | |
| | N776 | OD1 | D804 | OD2 | E779 | OE2 | E779 | OE1 | | | | |
| | D804 | OD2 | | | D804 | OD2 | D804 | OD1 | | | | |

crystal structure 3WGV but the value is 8.4 in chain C, yet, the structural difference between the two chains is minimal (*Table 2*). Because of this, only the structures with the highest resolution for the $E_1$ and $E_2$ states of the pump, 3WGV and 2ZXE, were used to guide the protonation state assignments in order to avoid false assignments of protonation state originated from structural uncertainty in lower resolution crystal structures. Interestingly, the protonation states of E327 and D804 assigned based on the 3WGV and 4HQJ structures are reversed. According to the PROPKA results, E327 appears to be deprotonated and D804 protonated in 3WGV, and vice versa in 4HQJ. While this inconsistency could be due to the lower resolution in the crystal structure of 4HQJ, it is worth noting that these two residues are in close proximity from one another suggesting that a proton could be passed back and forth between them in the $E_1$ state.

## Protonation states and $E_1$ binding pocket stability

Previous calculations have indicated that a specific set of residues has to be protonated for the $E_2$ binding sites to be $K^+$ selective (*Yu et al., 2011*). Although the protonation states of D926 and E954 were not considered in the previous study because they lie outside of the two $K^+$ binding sites, they are likely to be protonated in $E_2$ according to the predicted pKa (*Table 2*). The binding pocket

**Table 2.** pKa values of binding site titratable residues calculated from the crystal structures. The crystal structure resolution is given below the PDB ID.

| | $E_1$ | | $E_2$ | |
|---|---|---|---|---|
| | 3WGV<br>2.8 Å | 4HQJ<br>4.3 Å | 2ZXE<br>2.4 Å | 3B8E<br>3.5 Å |
| D804 | 5.9(6.2)* | 11.1 (11.2) | 3.7 | 0.8 (2.1) |
| D808 | 3.5(3.1) | 3.7 (3.7) | 5.8 | 6.8 (6.6) |
| D926 | 6.4(7.2) | 5.6 (5.6) | 8.9 | 7.4 (8.4) |
| E327 | 11.0(11.3) | 5.7 (5.6) | 8.3 | 10.8 (9.9) |
| E779 | 9.9(8.4) | 7.4 (7.3) | 10.7 | 9.6 (8.0) |
| E954 | 9.6(9.7) | 9.2 (9.2) | 10.3 | 10.7 (10.3) |

*If two chains are present in the same asymmetric unit, the pKa of the same residue in the other chain is shown inside the parenthesis.

protonation derived from the crystal structure of the pump in its partially occluded $Na_3 \cdot E_1(ADP \cdot Pi)$ state differs from that in $E_2$. According to their pKa values, D804, D808, and D926 should be deprotonated and the glutamates, E327, E786, and E954, should be protonated. Arguments based on pKa values predicted by PROPKA, however, have to be taken with caution, because the empirical method is highly sensitive to the local structural variation. For example, a deprotonated acidic side chain could have a PROPKA predicted pKa value at slightly higher than 7 because of the structural snapshot used to make the prediction. To seek a more robust assessment of these factors, molecular dynamics (MD) simulations were conducted to examine the structural stability of the binding pocket for different protonation configurations. One goal is to determine the protonation states of D804 and D926, both of which have a predicted pKa value within 1.5 pH unit to 7. The protonation state of D808 was also investigated since it affects the $K^+$ selectivity of the binding sites in the $E_2(K_2)$ state. A total of eight MD simulations were carried out (*Table 3*), starting from all possible protonation state configurations accessible by these three residues. The protocol for setting up these systems is given in details in the Materials and methods section.

*Figure 2* shows the binding pocket conformation in the $E_1$ systems (*Table 3*) at the end of the all-atom MD simulations. Out of the three aspartates when two or more are protonated the binding pocket becomes unstable, showing a large deviation from the crystal structure either with one of the bound $Na^+$ expelled from its binding site (systems E1_S5-7), or with a $Cl^-$ ion attracted into the binding pocket (system E1_S4). These scenarios are not likely to happen in the normal function cycle of the pump. On the other hand, when the number of protonated aspartates is less than two the binding pocket remains structurally close to that in the crystal structure (systems E1_S0-3). The heavy atom root mean squared deviations (RMSD) between the crystal structure and the structure snapshots from the last 50 ns of the simulations are less than 1.6 Å.

**Table 3.** Summary of the all-atom simulation systems and the FEP/H-REMD reduced systems. The binding site residues E327, E779, and E954 were kept protonated in all the systems.

| Systems | Binding site residues | | | Simulation time (ns) | |
|---|---|---|---|---|---|
| | | | | MD | FEP/H-REMD |
| Wildtype | | | | | |
| E1_S0 | D804- | D808- | D926- | 140 | 2 × 128 |
| E1_S1 | D804p | D808- | D926- | 300 | 2 × 128 |
| E1_S2 | D804- | D808p | D926- | 139 | 2 × 128 |
| E1_S3 | D804- | D808- | D926p | 503 | 2 × 128 |
| E1_S4 | D804p | D808p | D926- | 94 | |
| E1_S5 | D804p | D808- | D926p | 87 | |
| E1_S6 | D804- | D808p | D926p | 277 | |
| E1_S7 | D804p | D808p | D926p | 141 | |
| E2_S0 | D804- | D808p | D926- | 193 | 2 × 128 |
| E2_S1 | D804- | D808p | D926p | 350 | 2 × 128 |
| P-E2_S0 | D804- | D808p | D926p | 100 | 2 × 128 |
| Mutants | | | | | |
| E1_S1M | D804N | D808- | D926- | 40 | 2 × 128 |
| E1_S2M | D804- | D808N | D926- | 40 | 2 × 128 |
| E1_S3M | D804- | D808- | D926N | 40 | 2 × 128 |
| E2_S1M | D804N | D808- | D926p | 40 | 2 × 128 |
| E2_S2M | D804- | D808N | D926p | 40 | 2 × 128 |
| E2_S3M | D804- | D808- | D926N | 40 | 2 × 128 |
| P-E2_S1M | D804N | D808- | D926p | 40 | 2 × 128 |
| P-E2_S2M | D804- | D808N | D926p | 40 | 2 × 128 |

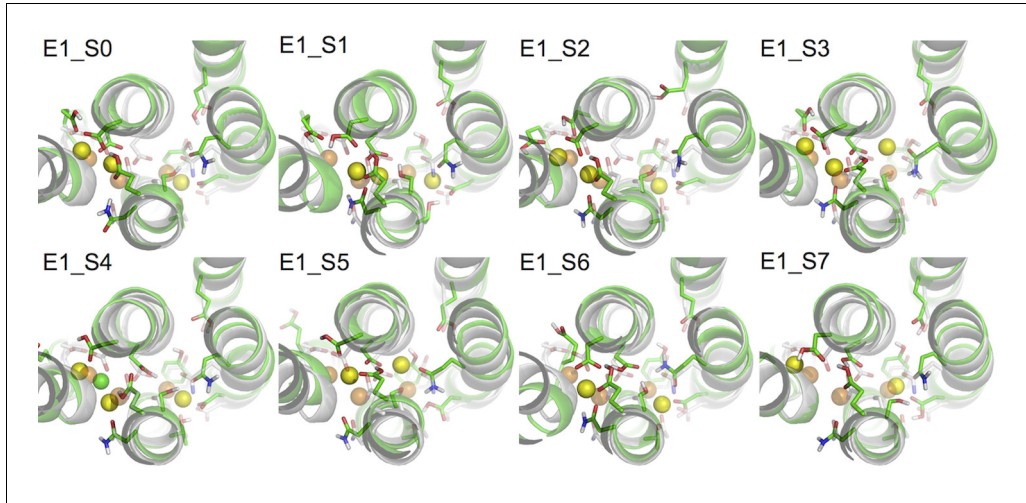

**Figure 2.** Comparison of snapshots at the end of the MD simulations (green) and the crystal structure of $Na_3 \cdot E_1 \cdot (ADP \cdot Pi)$ (PDBID 3WGV) (white). The binding site residues are shown in stick presentation and the ions are shown as spheres. Binding site $Na^+$ ions from the MD simulation snapshots are in yellow, and the crystal $Na^+$ are in orange. A $Cl^-$ ion has entered the binding site in system E1_S4 during the simulation and is shown in green. The view is from the extracellular side towards the intracellular side. The figure is produced with PyMOL (*DeLano, 2002*).

## Protonation state and the $E_1$ binding pocket selectivity

Analysis of the structural stability of the binding sites based on the MD simulations indicates that all of the four protonation states producing stable ion binding pockets may coexist when the pump is in the $Na_3 \cdot E_1 (ADP \cdot Pi)$ state. The relative population of these states in the state $Na_3 \cdot E_1 (ADP \cdot Pi)$, however, would differ from one another. The overall selectivity of the binding sites has contributions from all the states, and the predominant protonation state configuration should produce binding sites that are $Na^+$ selective. The determination of this protonation state configuration starts from the protein-membrane systems generated by the restraint-free MD simulations. A reduced structural model of the binding pocket is derived from each of these systems (*Table 3*) and the binding free energy differences between $Na^+$ and $K^+$ ($\Delta\Delta G_{Na \rightarrow K}$) at the binding sites are calculated (See Materials and methods). The value of $\Delta\Delta G_{Na \rightarrow K}$ reflects the affinity difference between $K^+$ and $Na^+$ binding as $\Delta\Delta G_{Na \rightarrow K} = \Delta G_k - \Delta G_{Na} = RT \ln\left(K_{D,K}/K_{D,Na}\right)$. The results are plotted as the logarithm of the affinity ratio, $\ln\left(K_{D,K}/K_{D,Na}\right)$, in *Figure 3*. The values of $\Delta\Delta G_{Na \rightarrow K}$ are shown in *Table 4*.

As shown in *Figure 3*, the protonation state of the aspartates greatly affects both the $E_1$ and $E_2$ binding sites selectivity. The protonation state at $E_2(K_2)$ binding pocket proposed in the earlier study (*Yu et al., 2011*) is confirmed by the present calculations, in which the reduced system includes a larger region of the protein and more residues at the outskirt of the main binding pocket, including D926 and E954. It is evident that a different set of residues is protonated in the $Na^+$ selective $E_1$ as compared to in the $K^+$ selective $E_2$ (*Figure 3*). While the glutamates (i.e., E327, E779, and E954) remain protonated in both the $Na_3 \cdot E_1 (ADP \cdot Pi)$ and $E_2(K_2)$ states, the binding pocket aspartates take on opposite protonation states. In $Na_3 \cdot E_1 (ADP \cdot Pi)$ D804 is protonated and both D808 and D926 are deprotonated, and their protonation states are reversed in $E_2(K_2)$. Free energy perturbation (FEP) calculations on the protonation/deprotonation of D804 estimated a pKa value of 9.2 (See Materials and methods), further supporting its protonated form under physiological conditions in $E_1$.

## Charge-neutralizing mutations and shift in selectivity

The impact of charge-neutralizing mutations of the key aspartate residues on the binding site selectivity was examined. D to N mutations have previously been used in experimental studies as a strategy to ascertain the possible charged state of these residues. The results of the free energy calculations, summarized in *Figure 4*, indicate that all the three sites remain $Na^+$ selective in the

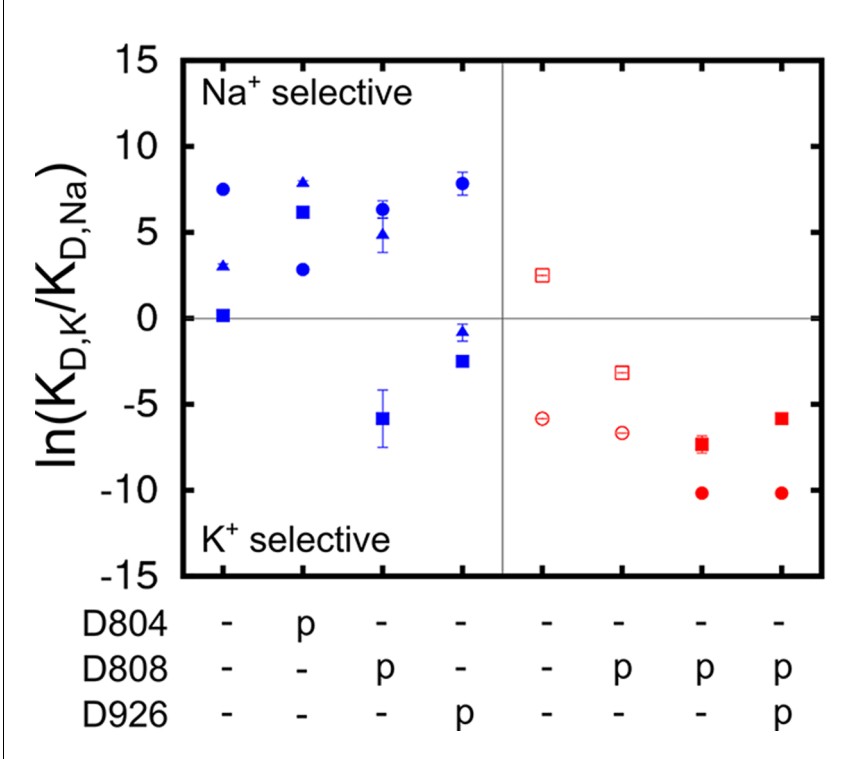

**Figure 3.** Ion binding sites selectivity characterized by $\ln(K_{D,K}/K_{D,Na})$ in states $Na_3E_1\cdot(ADP\cdot Pi)$ (blue) and $E_2(K_2)$ (red). Sites I (square), II (circle), and III (triangle) are distinguished by their shapes. Values from the previous calculations with a smaller reduced region are shown as empty symbols. All the binding site glutamates (i.e., E327, E779, and E954) are kept protonated. The protonation states of the binding site aspartates are indicated below the plot.

partly occluded $Na_3\cdot E_1(ADP\cdot Pi)$ in both D804N and D808N, whilst D926N causes the sites to lose most of the $Na^+$ selectivity. Using $Na^+$ dependent phosphorylation and ATP binding assays, it was shown that the cytoplasmic binding affinity for both $Na^+$ and $K^+$ is reduced in D804N and D808N (*Jorgensen et al., 2001*; *Pedersen et al., 1997*). The D804N mutation affects the cytoplasmic $K^+$ and $Na^+$ binding differently. The mutation's impact on cytoplasmic $Na^+$ binding is not as prominent as on $K^+$ binding and D804N would appear more $Na^+$ selective than the wildtype. This is reflected in the calculation results. The $K_{D,K}/K_{D,Na}$ at sites I and III are reduced by a factor of ~50 from 476.6 and 2523.3 in the wildtype to 10.3 and 54.6 in the D804N mutant, but the sites are still $Na^+$ selective. The selectivity of site II is increased to a much larger extent, from 17.0 in the wildtype to 13359.7 in the mutant. This is an ~1000 fold increase in selectivity and it implies that D804N is likely more $Na^+$ selective than the wildtype pump. The D808N mutant also becomes more $Na^+$ selective compared to the wildtype pump. The most dramatic increase in $Na^+$ selectivity happens at site II as in the case of D804N. Interestingly, site I, which prefers to bind $K^+$ in the wildtype with the protonated D808 (*Figure 3*), is now $Na^+$ selective in D808N. It could be that the spatial packing is the major contributing factor to the site I ion selectivity. In the calculations, both sites II and III lose their $Na^+$ selectivity in the D926N mutant. The shifting of selectivity towards $K^+$ at site III in this mutant is a bit surprising, and worth commenting on. There are two possible explanations. First, substituting a negatively charged carbonyl oxygen with a bulkier but neutral $-NH_2$ at the D926 sidechain could prevent entrance of an ion to this site. Thus, the simulated conformation with an ion at this site could be energetically inaccessible. In other words, this is a metastable state with the absolute free energy of $Na^+$ and $K^+$ binding to this conformation equally prohibitive. An alternative explanation is that the D926N mutation alters the available conformational space accessible by helix M5 and this allows $K^+$ to go into site III as suggested in reference (*Kanai et al., 2013*). The binding of this $K^+$, however,

**Table 4.** The binding free energy difference ($\Delta\Delta G_{Na \to K}$) at all the binding sites calculated from FEP/H-REMD simulations. The energy values are in kcal/mol.

| Systems | Binding sites | | |
|---|---|---|---|
| | I | II | III |
| Wildtype | | | |
| E1_S0 | 0.1 ± 0.1 | 4.5 ± 0.1 | 1.8 ± 0.1 |
| E1_S1 | 3.7 ± 0.2 | 1.7 ± 0.1 | 4.7 ± 0.1 |
| E1_S2 | -3.5 ± 1.0 | 3.8 ± 0.3 | 2.9 ± 0.6 |
| E1_S3 | -1.5 ± 0.2 | 4.7 ± 0.4 | -0.5 ± 0.3 |
| E2_S0 | -4.4 ± 0.3 | -6.1 ± 0.0 | |
| E2_S1 | -3.5 ± 0.1 | -6.1 ± 0.1 | |
| P-E2_S0 | -1.0 ± 0.6 | 0.7 ± 0.4 | |
| Mutants | | | |
| E1_S1M | 1.4 ± 0.1 | 5.7 ± 0.0 | 2.4 ± 0.1 |
| E1_S2M | 4.3 ± 0.7 | 6.6 ± 0.6 | 5.2 ± 0.9 |
| E1_S3M | -0.6 ± 0.1 | 1.7 ± 0.1 | 0.0 ± 0.0 |
| E2_S1M | -3.3 ± 0.3 | -5.7 ± 0.0 | |
| E2_S2M | -5.4 ± 0.1 | -6.4 ± 0.1 | |
| E2_S3M | -3.8 ± 0.1 | -7.2 ± 0.1 | |
| P-E2_S1M | 0.8 ± 0.9 | 1.9 ± 0.4 | |
| P-E2_S2M | 1.6 ± 0.5 | 1.9 ± 0.4 | |

prevents the further binding of $Na^+$, and results in the compromised $Na^+$ selectivity in this mutant. Even though the calculations show D926N with decreased $Na^+$ selectivity, they do not explain why experimentally the D926N mutant has compromised $Na^+$ binding ability in the absence of $K^+$ (*Einholm et al., 2010*). The loss in selectivity seen in this D926N mutant, however, could account for the strong inhibition by high $K^+$ on $Na^+/K^+$-ATPase activity (*cf.* **Figure 2A** in reference [*Einholm et al., 2010*]).

All the D to N mutations tested for the occluded $E_2(K_2)$ state appear to have little impact on the $K^+$ selectivity, in apparent contradiction with the experimental observations showing that both D804N and D808N display decreased selectivity for external $K^+$ (*Kuntzweiler et al., 1996*; *Pedersen et al., 1997*). In a previous computational study, the D808N mutation was also shown to minimally affect the $K^+$ selectivity in $E_2(K_2)$. Discrepancies between the calculations and the selectivity inferred from biochemical experiments have been noted previously, though the reason was not clear. A plausible explanation could be that the experimentally measured selectivity is an outcome from the relevant states including both P-$E_2$ and $E_2(K_2)$. However, any contribution from the P-$E_2$ state to the observed selectivity was not taken into account by the calculations because of the missing crystal structure of the outward-facing state. To test this idea, we generated a model structure of the P-$E_2$ state based on the homolog structures from the $Ca^{2+}$ SERCA pump (see below). This model was then used to calculate the $\Delta\Delta G_{Na \to K}$ in the wildtype and the mutant pumps. The results show that both sites I and II have increased preference for external $Na^+$ in the P-$E_2$ state of D808N (*Figure 4*), which explains previous discrepancies between the calculations and the experiments.

## Ion selectivity along the pump cycle

Using the crystal structures of the $Ca^{2+}$ SERCA pump as templates and a coarse grained transition pathway calculation method, ANMpathway (*Das et al., 2014*), we generated structural models of the $Na^+$– and $K^+$–loaded outward-facing P-$E_2$ state (see Materials and methods). The models appear to be structurally similar to the recently published P-$E_2$ like structures of the $Na^+/K^+$-pump (*Gregersen et al., 2016*; *Laursen et al., 2013*) and the vanadate-Inhibited, P-$E_2$ mimic of the $Ca^{2+}$

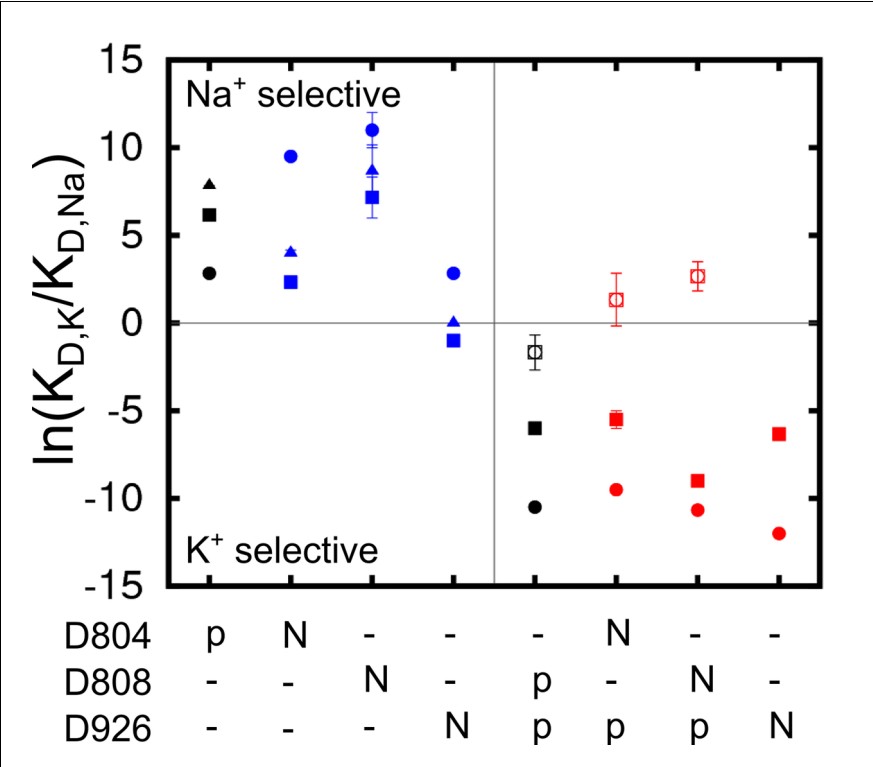

**Figure 4.** Charge-neutralizing mutations and their impact on binding site selectivity. The wildtype protein is colored black and the mutations in $E_1$ (blue) and $E_2$ (red) are colored differently. Sites I (square), II (circle), and III (triangle) are distinguished by their shapes. The empty symbols represent values calculated from the outward facing P-$E_2$ model.

SERCA pump (*Clausen et al., 2016*). The MD simulations of the models are also able to predict with considerable accuracy the experimentally measured gating charge upon ion binding (*Castillo et al., 2015*). The structural transition pathways leading to these models provide a view of the intermediate structures along the pump cycle. Using these models, we calculated the $\Delta\Delta G_{Na \to K}$ for the binding site of the intermediate state structures. The entire atomistic protein-membrane systems were used in the calculations. The results are presented in *Figure 5* and *Table 5*. The calculations are valid as the $\Delta\Delta G_{Na \to K}$ values mirror those computed from the reduced systems with the same protonation states (*Figure 3*).

The results reveal a fascinating feature of the binding site selectivity along the pump cycle. Initially, the two $K^+$ binding sites in the outward-facing model P-$E_2 \cdot K_2$ (state ① in *Figure 5*) appear to be non-selective or only weakly $K^+$ selective. However, the selectivity for $K^+$ over $Na^+$ increases as the pump occludes to form the $K^+$–bound occluded state $E_2(K_2)$ (state ③). After the dephosphorylation of P-$E_2$, the binding pocket opens up to the cytoplasmic side. Accompanying this structural transition is the selectivity reversal at site I, switching the site from $K^+$ to $Na^+$ selective (③ to ④ in *Figure 5*). The protonation state change further shifts the ion selectivity at the sites in $E_1$ towards $Na^+$. Changing the protonation state in $K_2E_1$ (④ to ⑤ in *Figure 5*) to the protonation state dominant in $Na_3 \cdot E_1$(ADP·Pi), i.e., state ⑥ in *Figure 5* with protonated D804 and deprotonated D808 and D926, further reduces the $K^+$ selectivity at both sites I and II. When the pump is in this state, site I is $Na^+$ selective and site II is only weakly $K^+$ selective. Together, the $E_2$ to $E_1$ structural transition and the protonation state change promote the release of $K^+$ to the cytoplasmic side. Transient changes in the binding pocket protonation state upon occlusion/deocclusion are possible and could lead to variations in the magnitude of $\Delta\Delta G_{Na \to K}$ for the intermediate states. Nonetheless, the general trend in free energy changes upon occlusion/deocclusion should remain because the $\Delta\Delta G_{Na \to K}$ of the

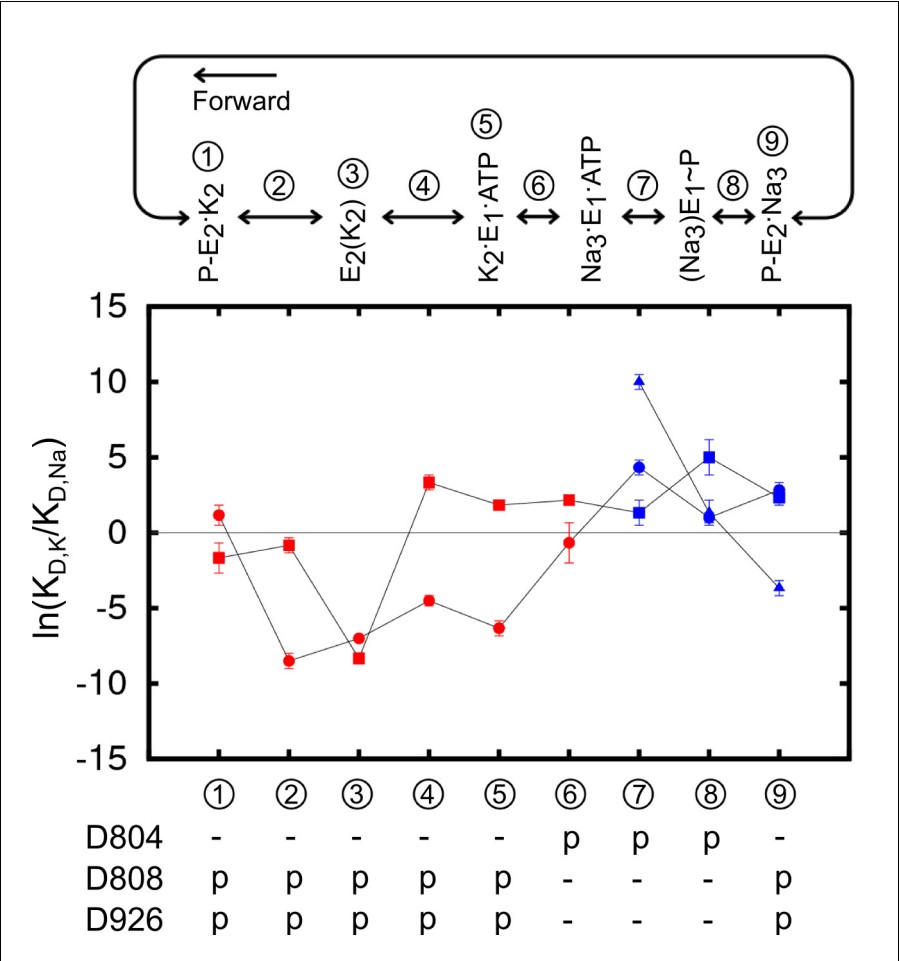

**Figure 5.** The binding site ion selectivity along the pump cycle. Sites I (square), II (circle), and III (triangle) are distinguished by their shapes. Different colors indicate whether there are two (red) or three (blue) sites that are included in the calculations. The conformational states are numbered and stamped along the pump cycle in the top panel. The protonation states of the aspartates are indicated below.

occluded state $E_2(K_2)$ has a much larger magnitude than that of the open-access outward- and inward-facing states.

The free energy calculations corroborate the notion that the two $K^+$ binding from the extracellular side is sequential and possibly cooperative. The sequential binding of extracellular $K^+$ was initially demonstrated by Forbush (*Forbush, 1987*). Recently using crystallography and isotopic measurements Ogawa *et al.* presented strong evidence that the first $K^+$ binds at site I and the second $K^+$ at site II (*Ogawa et al., 2015*). Whether there is cooperativity upon extracellular $K^+$ binding, however, is not clear from the crystal structures. Two electrode voltage clamp experiments have shown that the two extracellular $K^+$ binding events are relatively independent in the absence of extracellular $Na^+$, but there is positive cooperativity of $K^+$ binding when extracellular $Na^+$ ions are present (*Jaisser et al., 1994*). The current $\Delta\Delta G_{Na\to K}$ calculations of the two ion bound P-$E_2$ (*Table 5*) provide an interpretation of this phenomenon. With a $K^+$ ion at site I, site II is relatively nondiscriminatory ($\Delta\Delta G_{Na\to K}$ = 0.7 kcal/mol, p/p to p/s in state ① P-$E_2$.$K_2$, *Table 5*), but it becomes much more $Na^+$ selective ($\Delta\Delta G_{Na\to K}$ = 2.4 kcal/mol, s/s to s/p in state ⑨ P-$E_2$·$Na_3$, *Table 5*) when there is a $Na^+$ ion bound at site I. In the presence of extracellular $Na^+$, $K^+$ ions have to first compete with $Na^+$ to bind at site I in order for the subsequent $K^+$ to bind, therefore resulting in the observed binding cooperativity in the presence of extracellular $Na^+$.

**Table 5.** Selectivity in the form of at the binding sites along the pump cycle from state P-E$_2$·K$_2$ to P-E$_2$·Na$_3$. The energies are in kcal/mol.

| | *p/- | -/p | p/p | p/p | p/p |
| | s/- | -/s | s/p | p/s | s/s |
|---|---|---|---|---|---|
| (1) P-E$_2$.K$_2$ | -0.7 ± 1.0 | 0.1 ± 0.5 | -1.0 ± 0.6 | 0.7 ± 0.4 | -0.6 ± 0.7 |
| (2) Intermediate | 0.0 ± 0.4 | 1.8 ± 1.3 | -0.5 ± 0.3 | -5.1 ± 0.3 | -4.0 ± 0.9 |
| (3) E$_2$(K$_2$) | -4.9 ± 0.8 | -3.0 ± 0.5 | -5.0 ± 0.2 | -4.2 ± 0.1 | -9.3 ± 0.5 |
| (4) Intermediate | 0.6 ± 0.7 | -0.8 ± 0.3 | 2.0 ± 0.3 | -2.7 ± 0.2 | -1.6 ± 0.4 |
| (5) K$_2$.E$_1$ | 1.5 ± 0.3 | 1.1 ± 0.2 | 1.1 ± 0.1 | -3.8 ± 0.3 | -1.2 ± 0.5 |
| (6) K$_2$.E$_1$* | 0.6 ± 0.5 | 0.7 ± 0.4 | 1.3 ± 0.2 | -0.4 ± 0.8 | 0.1 ± 0.7 |

| | *s/-/- | -/s/- | s/s/- | s/s/- | s/s/- | s/s/s | s/s/s | s/s/s | s/s/s | s/s/s | s/s/s | s/s/s |
| | p/-/- | -/p/- | p/s/- | s/p/- | p/p/- | p/s/s | s/p/s | s/s/p | p/p/s | p/s/p | s/p/p | p/p/p |
|---|---|---|---|---|---|---|---|---|---|---|---|---|
| (7) Na$_3$·E$_1$(ADP·Pi) | -1.0 ± 0.5 | 1.7 ± 0.2 | 0.7 ± 0.7 | 2.2 ± 0.2 | 3.0 ± 1.0 | 0.8 ± 0.5 | 2.6 ± 0.3 | 6.0 ± 0.3 | 2.1 ± 1.5 | 5.8 ± 1.1 | 7.4 ± 0.8 | 7.5 ± 2.0 |
| (8) Intermediate | 2.9 ± 1.0 | -0.5 ± 1.2 | 1.5 ± 1.6 | 2.6 ± 1.3 | 2.5 ± 1.3 | 3.0 ± 0.7 | 0.6 ± 0.2 | 0.8 ± 0.5 | 4.0 ± 0.8 | 2.8 ± 1.1 | 2.7 ± 0.5 | 3.5 ± 1.5 |
| (9) P-E$_2$·Na$_3$ | 2.3 ± 0.3 | 1.6 ± 0.5 | 2.1 ± 0.3 | 2.4 ± 0.3 | 4.6 ± 0.5 | 1.4 ± 0.3 | 1.7 ± 0.3 | -2.2 ± 0.3 | 3.1 ± 0.8 | -0.6 ± 0.6 | -0.8 ± 0.6 | -1.2 ± 1.2 |

*The top and bottom rows represent the starting and ending binding site ion configurations. A 'p' represents a K$^+$ (potassium) ion and an 's' represents a Na$^+$ (sodium) ion. The binding sites I, II, and III in this order are separated by '/'.

A similar phenomenon is seen in the Na$^+$ branch of the pump cycle. Initially, site I in the inward-facing Na$_3$·E$_1$(ADP·Pi) structure (state ⑦) is weakly Na$^+$ selective when the other two sites are filled with Na$^+$. However, the intermediate state ⑧ during the occlusion process shows that its site I is highly discriminating against K$^+$ with a $\Delta\Delta G_{Na \to K}$ of 2.9 kcal/mol (*Figure 5* and *Table 5*). Therefore, if a K$^+$ is bound in this site, the free energy barrier toward occlusion would be much higher than when a Na$^+$ ion was bound and the subsequent occlusion is not likely. In the case when a K$^+$ ion replaces a Na$^+$ at site II or III, although the energy barrier of occlusion is not as prominent, such a binding pocket ion configuration is energetically less favorable than the native configuration and its appearance is unlikely in the first place.

Unlike the K$^+$ branch, the $\Delta\Delta G_{Na \to K}$ calculations along the Na$^+$ branch offers limited insights on the sequence of Na$^+$ binding from the cytoplasmic side. The results are indicative of the selectivity at the sites, but not the absolute binding affinity. Based on the calculations alone, it is not possible to determine which site is the first to bind cytoplasmic Na$^+$. It could be site III, as suggested by Kanai et al., and this prepares the other two sites for the following Na$^+$ binding (*Kanai et al., 2013*). Or, alternatively, the first two Na$^+$ bind at sites I and II in the main binding pocket and the last Na$^+$ enters, takes over site II, and pushes the two previously bound ions further, so that the ions in sites II and I now move to sites I and III, in a process reminiscent of the "knock-on" mechanism occurring in potassium channels (*Hodgkin et al., 1955*). A direct assessment of these proposed binding sequences will require further experiments.

## Discussion

The results from the MD simulations support the notion that the protonation state of the binding pocket and its selectivity are closely related. Because the binding pocket in the Na$^+$/K$^+$-pump displays considerable flexibility, it is worth pausing to reflect on the possible mechanism that underlies this relation. While the selectivity of a very rigid binding site is first and foremost predetermined by its atomic geometry, the selectivity of a flexible binding site is strongly affected by local ion-ligand and ligand-ligand interactions. In such flexible systems, selectivity is controlled by the both number and the physicochemical properties of ion-coordinating ligands (*Yu et al., 2010*). For example, high-field ligands, such as deprotonated acidic side chains tend to favor Na$^+$ binding and protonation revert those to low-field ligands, which tend to favor K$^+$. Hence, in the Na$^+$/K$^+$-pump protonation is exploited to modulate selectivity by altering the electrostatic properties of several of residues in the binding pocket. This is also consistent with the results from our previous study on the ion selectivity

in $E_2(K_2)$, which concluded that changes in the electrostatic properties of the protonatable residues was the likely mechanism responsible for the $K^+$ selectivity in the $E_2$ state of the pump (*Yu et al., 2011*). Even though the local structural rearrangements at the binding sites are small upon the change in protonation state, it is enough to change the physical properties of the coordinating ligand, thus giving rise to discernable differences in the ion selectivity. The crystal structures of the $Na^+/K^+$-pump in its $Na_3 \cdot E_1 \cdot (ADP \cdot Pi)$ and $E_2(K_2)$ states show similarity in their binding pocket configurations (*Figure 1*), including the coordination patterns of the bound ions at sites I and II (*Table 1*). The heavy atom RMSD between the binding pocket residues is 2.5 Å and the structural difference remains after hundreds of ns of simulations. Empirical pKa and FEP calculations based on the MD simulation equilibrated structures indicate that the binding pocket glutamates (i.e., E327, E779, and E954) are likely protonated in both $Na_3 \cdot E_1 \cdot (ADP \cdot Pi)$ and $E_2(K_2)$, although previous mutagenesis experiments showed that charge-neutralizing mutation E327Q affects pump function, possibly by altering ion-pump interactions and the kinetics of the occlusion/deocclusion reactions along the pump cycle (*Jorgensen et al., 2001*; *Kuntzweiler et al., 1995*; *Li et al., 2006*; *Nielsen et al., 1998*). The calculations also suggest that the protonation states of D804, D808, and D926 are different in $Na_3 \cdot E_1 \cdot (ADP \cdot Pi)$ and $E_2(K_2)$. Among the three aspartates only D804 is protonated in order for all three sites to stay $Na^+$ selective in $Na_3 \cdot E_1 \cdot (ADP \cdot Pi)$. The $E_1$ binding pocket devoid of ions carries a net charge of -2, consistent with that deduced from previous fluorescence studies (*Schneeberger et al., 1999*). Mutagenesis experiments are also consistent with the unlikely protonation of D808 and D926 when the pump is in $E_1$ trying to bind cytoplasmic $Na^+$ (*Jewell-Motz et al., 1993*; *Pedersen et al., 1997*). The protonation states of these three residues are reversed in $E_2(K_2)$ with D804 deprotonated and the other two protonated, a result that is supported by a previous computational study (*Yu et al., 2011*). The different protonation states for $E_1$ and $E_2$ also agree well with the 'four-site' model proposed by Skou and Esmann more than 30 years ago with the $K^+$-bound $E_2$ state carrying two sidechain protons ($H_2EK_2$) and the $Na^+$-bound $E_1$ state carrying only one ($HENa_3$) (*Skou et al., 1980*).

It is worth pointing out that the second proton (i.e., the proton on D926) in the $K^+$ bound $E_2$ state must come from and return to the same side of the membrane during the pump cycle so that the net charge moved per cycle by the pump is one. It seems unlikely that this proton could come from the extracellular side because altering extracellular pH would then protonate or deprotonate D926, causing major changes in $Na^+$ binding affinity and the maximum pumping turnover rate. This is not observed over an external pH range 9.6 to 5.6 (*Mitchell et al., 2014*; *Vasilyev et al., 2004*; *Yu et al., 2011*). Therefore, it is more likely that the D926 proton comes from the cytoplasmic side. This is supported by MD simulations of the $P-E_2 \cdot Na_2$ and $Na_2 \cdot E_1 \cdot (ADP \cdot Pi)$ revealing the existence of aqueous pathways connecting the cytoplasm and D926 (*Figure 6*). One of the water pathways is located between the helices M5, M7, and M8 (*Figure 6B*), similar to the C-terminal proton pathway previously proposed (*Poulsen et al., 2010*). A proton could enter through this pathway to protonate D926 and then leave through the same pathway during the $E_2$ to $E_1$ transition, or through an alternative path passing the main binding pocket along with the dissociating ions as seen in the $Na_2 \cdot E_1 \cdot (ADP \cdot Pi)$ simulation (*Figure 6A*).

Under physiological conditions, the challenge faced by the $Na^+/K^+$-pump is to effectively pick out the correct ion species from a solution much more concentrated with other types of ions. Even when the binding affinity of the ion species being transported is a few orders of magnitude higher than that of the other ion ($\Delta\Delta G_{bind}$ = 1 to 3 kcal/mol), the advantage in binding is undermined by the higher concentration of the competing ion. The pump overcomes this problem by raising the free energy barrier for the occlusion step in the presence of incorrect ions in the binding pocket, thus preventing the faulty transport (*Figure 5* and *Table 5*). This self-correcting mechanism makes sure that the intracellular and the extracellular gates do not close (i.e., occlude), which is required for the pump cycle to go forward, unless the correct ion configuration is present at the binding pocket. Therefore, this is an inherent part of the ping-pong mechanism the pump uses to transport ions with high selectivity. Although other energetic barriers, like pathways the ions use to travel to the binding site, could contribute to the ion binding specificity, it is not likely the case here as both $Na^+$ and $K^+$ can access the binding pocket in the simulations of the $E_1$ and $P-E_2$ states without ions bound.

The concept of a self-correcting pumping mechanism sheds new light on a number of puzzling functional observations. For instance, it explains why $E_1$ displays a strong apparent affinity for $K^+$ in competitive binding assays (*Schneeberger et al., 2001*), even though the pump still binds and

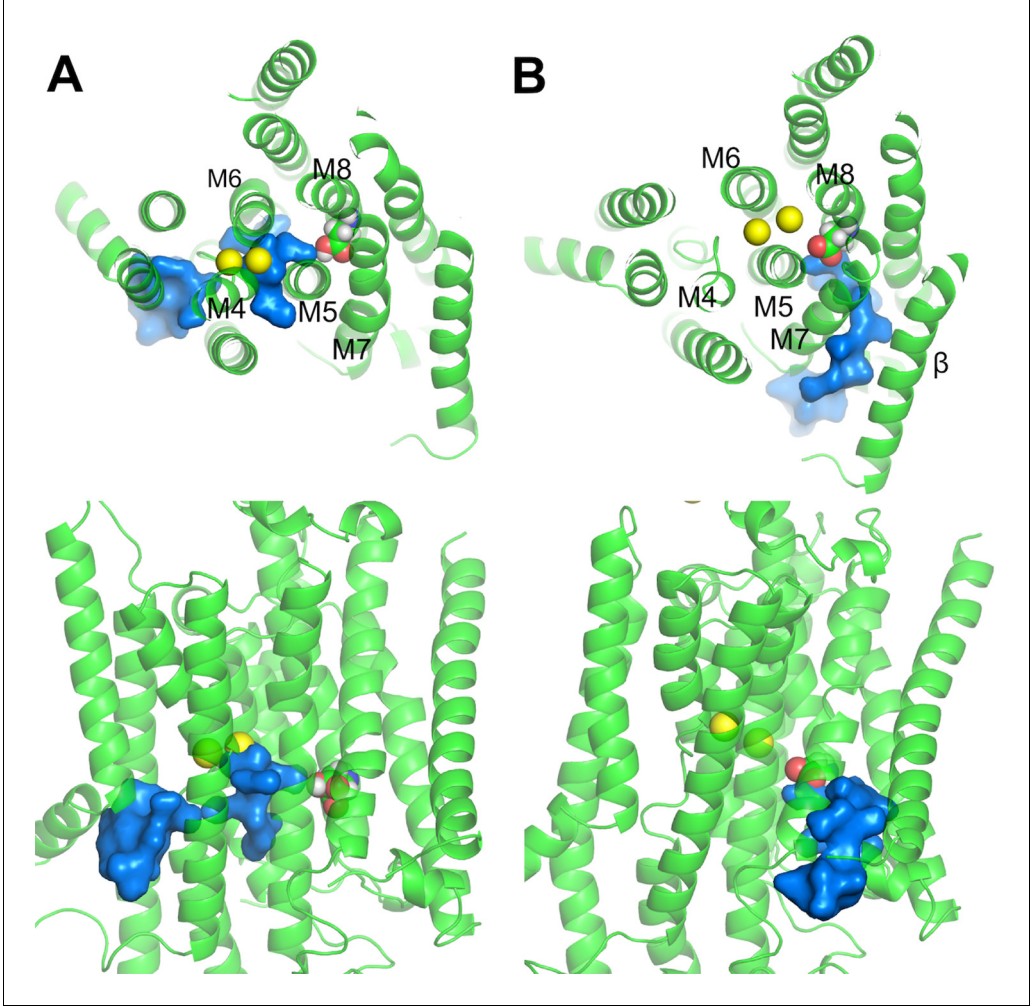

**Figure 6.** Water pathways from the cytoplasm to D926 in (**A**) $Na_2 \cdot E_1 \cdot (ADP \cdot Pi)$ and (**B**) $P-E_2 \cdot Na_2$. The top (top) and side (bottom) views are shown. D926 are shown in sphere representation. Water path connecting the cytoplasm and the D926 are in surface representation colored in blue. $Na^+$ (yellow) are shown as spheres.

occludes $Na^+$ from the cytoplasmic side. The calculations show that all the sites in the $Na_3 \cdot E_1 \cdot (ADP \cdot Pi)$ state are selective for $Na^+$, likely because this is a partially occluded state. A wrong combination of binding site ions would not have been occluded and reached such a state. In a fully inward-open conformation, the selectivity of the sites ought to be fairly weak. The high affinity for $K^+$ observed in $E_1$ in these experiments is due to the backward occlusion leading to the $K^+$-bound state $E_2(K_2)$. The proposed self-correcting mechanism parallels the suggestion based on experiments that the phosphorylation and occlusion of $E_1$ requires 3 $Na^+$ bound, and this increases the apparent affinity for $Na^+$ in the normal pump cycle (*Schneeberger et al., 2001*). The self-correcting mechanism can also account for the different effects of $Na^+$ and $K^+$ congeners on the release rate of the occluded $^{86}Rb^+$ (a $K^+$ congener) to the extracellular side upon the "backdoor" phosphorylation in the presence of Pi reported by Forbush (*Forbush, 1987*). This backward deocclusion is accelerated in the presence of $Na^+$, and it follows a single phase, while a much slower deocclusion process with a second slow phase is observed in the presence of $K^+$ or $Rb^+$. According to the self-correcting mechanism, it is likely that upon the replacement of one of the two occluded $^{42}K^+$ or $^{86}Rb^+$ by cold $K^+$ at one site, the pump succeeded to occlude again, resulting in the second slow phase. These explanations are sensible, in turn lending support to the proposed self-correcting mechanism. Given that the selectivity in the outward/inward facing states is after all not as strong as previously thought, such a mechanism must be in place.

This analysis suggests that an important component of the overall selectivity emerges from the increased free energy barrier associated with the occlusion process while the wrong type of ion is bound to a site. Although the pump would be kinetically more efficient by preventing the wrong ions from reaching the binding sites to begin with, the free energy calculations do not support the notion of highly selective binding sites for the open-access states. While such mechanism may seem inefficient because the pump must try to figure out that there is an issue with selectivity only after binding of several ions of the wrong type, it is important to realize that the $Na^+/K^+$-pump is not a particular fast molecular machine. The estimated turnover rate is less than 100 per second and decreases even further as the transmembrane voltage becomes more negative (*Heyse et al., 1994*). The implication is that the system has plenty of time, at the molecular level, to function near thermo-dynamic equilibrium. In other words, the pump has not been evolutionarily optimized to be a particularly fast molecular machine, but an energetically efficient one. Thus, even though the idea of a self-correcting ion selectivity mechanism seems counterintuitive and inefficient, it is consistent with the physical conditions under which the pump has to operate.

In summary, the present study highlights the tight coupling between the $Na^+/K^+$ selectivity of the binding sites, the protonation state of the coordinating residues, and the conformational state of the pump. The important functional consequence of such tight coupling is the necessity to have the correct type and number of ions in the binding pocket for the pumping cycle to proceed forward toward the occlusion step. Because the ion binding selectivity is strongly dependent upon the protein conformation, a self-correcting mechanism counteracting the effect of the ion concentration in the environment ensures an efficient function of the pump.

## Materials and methods

### Preparing the all-atom membrane-$Na^+/K^+$-pump simulation systems

The crystal structures, 3WGV (*Kanai et al., 2013*) and 2ZXE (*Shinoda et al., 2009*), representing the $Na^+/K^+$-pump in its $Na_3 \cdot E_1(ADP \cdot Pi)$ and $E_2(K_2)$ states were used to build the simulation systems. The structure 3WGV contains two copies of the pump assembly in the asymmetric unit. Since the structural variations between the two assemblies are minimal, only the copy including chains A, B, and G was kept. Several small molecule ligands were co-crystalized with the pump in 3WGV, including an ADP, a $AlF_4^-$ ion, an oligomycin A, two $Mg^{2+}$, three cholesterol, four $Na^+$, and five 1,2-diacyl-sn-glyc-ero-3-phosphocholine molecules for each copy of the pump structure. Among these ligands, oligo-mycin A was not included in the simulation system. The ion was replaced by a $PO_4^{3-}$ and POPC molecules were used in place of the 1,2-diacyl-sn-glycero-3-phosphocholine. The structure of 2ZXE also contains small molecule ligands including cholesterol, $K^+$, $Mg^{2+}$, and $MgF_4^{2-}$. Similarly, a $PO_4^{3-}$ was used to replace the $MgF_4^{2-}$ ion. Since the $Na^+/K^+$-pump crystal structures were obtained from different organisms, the residue numbers differ slightly. For the sake of convenience, the numbering scheme in the newly resolved 3WGV from pig kidney was adopted in this study.

After removing irregularities from the PDB files, the $Na^+/K^+$-pump subunits were capped with acetylated N-termini and amidated C-termini. The ectodomain of the β-subunit was not included in simulation systems to reduce the computational cost. The orientation was chosen to be the same as in the OPM database (*Lomize et al., 2006*). At this stage, the protonation states of the binding site residues were assigned with the PATCH command in CHARMM (*Brooks et al., 2009*). Eight different protonation states were considered in $Na_3 \cdot E_1(ADP \cdot Pi)$ state systems and both the protonated and charged D926 were included in the $E_2(K_2)$ state systems. This resulted in a total of ten systems before proceeding to the next step. *Table 3* shows the system names and the associated binding site protonation. We used the *membrane builder* module in CHARMM-GUI (*Jo et al., 2007*, *2008*, *2009*; *Wu et al., 2014*) to generate POPC bilayers around the pump structures. Once this was completed, each protein-membrane complex was solvated by an equal molar mixture of KCl and NaCl. The final system had a combined cation concentration of 0.3 M (i.e., $[K^+] = [Na^+] = 0.15$ M). At the end the E1 system was $84 \times 110 \times 158$ Å$^3$ in size and contained ~138,000 atoms, while the dimension of the E2 system was $85 \times 109 \times 105$ Å$^3$ with ~134,000 atoms. Each completed system was subjected to a 675-ps equilibration with reducing restraints on the heavy atoms to relax the initially uncorrelated components, followed by a 10-ns unrestrained pre-production using the simulation package NAMD2.9 (*Phillips et al., 2005*). After the systems were well equilibrated, they were

simulated longer using the special-purpose supercomputer Anton (*Shaw et al., 2009*), which is designed for long time scale MD simulations.

## Generating the D to N mutant systems

Experimentally asparagine and glutamine are used as surrogates for protonated aspartate to study the effect of protonation. The effects of these charge-neutralizing mutants on the selectivity of the pump can be investigated computationally. The most interesting mutations in the context of this study are the single D to N mutations at the binding pocket, including residues D804, D808, and D926. The mutations were made on the crystal structures by replacing the proton on the OD2 atom in the aspartate with an –$NH_2$ amine group. D804N and D808N mutations were also made in the outward facing P-$E_2$ structural model (see below) to study how they affect the external $K^+$ binding. The protonation states of the other titratable residues were determined with PROPKA. Each of these systems was equilibrated without any restraints for 40 ns. The system snapshot with the least structural deviation of the pump to the averaged structure during the simulation was used to generate the reduced system. The mutant systems are shown in *Table 3*.

## Selectivity at the binding site

The absolute free energy of an ion *i* binding to a binding site inside a protein has the following form,

$$\Delta G_{i,\,\text{bind}} = \left( G_{i,\,\text{int}}^{\text{site}} - G_{i,\,\text{int}}^{\text{bulk}} \right) + \left[ -k_{\text{B}}T\text{In}(F_t\text{C}^{\circ}) - G_{i,\,\text{trans}}^{\text{site}} \right]. \tag{1}$$

The difference in the first term, $G_{i,\,\text{int}}^{\text{site}} - G_{i,\,\text{int}}^{\text{bulk}}$, represents the nonbonded interaction component of the free energy change upon moving the ion from the bulk solution to the binding site. The subtraction in the second term, $-k_{\text{B}}T\text{In}(F_t\text{C}^{\circ}) - G_{i,\,\text{trans}}^{\text{site}}$, reflects the lost of translational freedom. The translational freedom factor $F_t$ in bulk solution can be evaluated analytically under a rigid rotor approximation (*Deng et al., 2006*). Its final form depends on the force constants and the equilibrium values in the distance and angle restraints applied on the ion and the surrounding protein atoms. Based on *Equation (1)*, the selectivity of a binding site can be expressed as the binding free energy difference of two ion species. For example, the binding free energy change upon changing ion *i* to *j* at the binding site is

$$\begin{aligned}\Delta\Delta G_{i\to j} &= \left( G_{j,\,\text{int}}^{\text{site}} - G_{j,\,\text{int}}^{\text{bulk}} \right) - \left( G_{i,\,\text{int}}^{\text{site}} - G_{i,\,\text{int}}^{\text{bulk}} \right) + \left( G_{i,\,\text{trans}}^{\text{site}} - G_{j,\,\text{trans}}^{\text{site}} \right) \\ &= \Delta G_{i\to j}^{\text{site}} - G_{i\to j}^{\text{bulk}} - G_{i\to j}^{\text{trans}}.\end{aligned} \tag{2}$$

A negative value of $\Delta\Delta G_{i\to j}$ indicates that ion *j* binds more favorably than ion *i* and the site is *j* selective. There are three terms to be evaluated in *Equation (2)*. $\Delta G_{i\to j}^{\text{site}}$ and $\Delta G_{i\to j}^{\text{trans}}$ are calculated using the reduce binding site model, while $\Delta G_{i\to j}^{\text{bulk}}$ is computed using a water sphere with the impact from the bulk solution factored in with a boundary potential (see below).

## Generating the reduced binding site systems

To generate the reduced binding site, the all-atom system prepared according to the procedures above was divided into an inner region and an outer region. The inner region was defined as residues and water molecules within 15 Å to the center of mass of the bound ions. Everything within this region was treated explicitly. An extended inner region was specified by a 3-Å thick shell continuing from the inner region outwards to create a smooth spherical dielectric cavity. Water molecules in this region were removed and their impact on the binding site was included implicitly. The coordinates of atoms in these extended region and those linked to them within three atomic bonds in the inner region were held fixed during the simulations. The rest of the system was considered the outer region, in which the atoms were removed and their impact on the inner region atoms was represented by the General Solvent Boundary Potential (GSBP) in the form of a solvent-shielded static field and a solvent-induced reaction field (*Im et al., 2001*). The reaction field arising from changes in charge distribution in the inner region was expressed in terms of a generalized multipole expansion using 11 spherical harmonic functions. Both the solvent-shielded static field and the reaction field matrix were independent of the inner region configuration, and therefore were calculated only once before further simulations using the finite-difference Poisson−Boltzmann (PB) method with the

PBEQ module (*Im et al., 1998*) in CHARMM. In these calculations a dielectric constant of 1 was used for the inside of the protein within the inner and outer regions, whereas the rest of the system had a dielectric constant of 80. The atomic Born radii used in the PB calculations were determined by free energy calculations in explicit solvent (*Nina et al., 1997*). All these reduced systems were further equilibrated for 200 ps using Langevin dynamics at 303.15 K with a friction coefficient of 5 ps$^{-1}$ assigned to all non-hydrogen atoms. All bonds involving hydrogen atoms were fixed with the SHAKE algorithm (*Ryckaert et al., 1977*). Nonbonded interactions within 14.5 Å were accounted for explicitly, while everything else beyond this distance was treated with an extended electrostatics method (*Stote et al., 1991*). The simulation program CHARMM (*Brooks et al., 2009*) was used for the equilibration. *Table 3* lists all the reduced systems.

## Free energy perturbation simulations with boosting potential

It is known that the binding free energy calculated from FEP/MD simulations suffers from convergence issues because the residue sidechains at the binding site only sample a few of the several accessible rotameric states. This problem can be alleviated by introducing a replica-exchange scheme aiming at enhancing the sampling of sidechains (*Jiang et al., 2009, 2010*). This scheme allows exchange between the neighboring λ windows. Each λ window is coupled with a set of replica systems with a boosting potential of increasing strength used to accelerate the inter-conversion between sidechain rotameric states. We employed this hybrid FEP/H-REMD scheme implemented in the REPDSTR module in CHARMM (*Jiang et al., 2010*) to calculate the $\Delta\Delta G_{i\rightarrow j}$.

The boosting potential for the $\chi_1$ sidechain torsion of a binding site residue was obtained by fitting the potential of mean force as a function of the torsion $\chi$, $\mathcal{W}(\chi)$, with a cosine Fourier series in the form of

$$U_{\mathrm{BP}}(\chi) = \sum_{n=1}^{n} K_n \left\{ 1 + \cos\left[ \mathrm{n}\left( \chi - \chi_{0,n} \right) \right] \right\}. \tag{3}$$

The angle $\chi_1$ is dihedral formed by the bonded atoms N, CA, CB, and CG. The total number of the cosine terms, N, varies from 3 to 6, depending on which one produce a better fit to the $\mathcal{W}(\chi)$. A residue is considered in the binding site if any of its sidechain heavy atoms is within 4.5 Å to the bound Na$^+$ or K$^+$. *Table 6* lists all the binding site residues included in the boosting potential calculations.

We performed umbrella sampling simulations to obtain the $\mathcal{W}(\chi)$ for each binding site residue. First, the entire transmembrane helix containing the residue of interest was taken from the crystal structure with its orientation kept the same as in the OPM database. This was to provide a proper secondary structure environment. Next, we replaced the sidechains of all the other residues on the helix to hydrogen atoms to remove their steric effects. An implicit planar membrane model, EEF1/IMM1 (*Lazaridis, 2003*; *Schneeberger et al., 1999*) was used in place of a membrane bilayer to solvate the helix. 72 umbrella windows along $\chi$ were set up with a window spacing of of 5°. A harmonic bias potential was applied in each window with a force constant of 100 kcal/mol/rad$^2$. After the windows were generated, each of them was simulated for 50 ns at 303.15 K using Langevin dynamics with a 2 fs time-step. All bond lengths involving hydrogen were fixed with the SHAKE algorithm (*Ryckaert et al., 1977*). The cutoff distance for nonbonded interactions was set to 11 Å. The simulation program CHARMM (*Brooks et al., 2009*) was used to conduct the simulations. The resulting distributions of $\chi$ were unbiased using the weighted histogram analysis method (WHAM) (*Kumar et al., 1992*). *Figure 7* shows the $\mathcal{W}(\chi)$ and the fitted curves. The fitting parameters and are given in *Table 6*. An appropriate boosting potential can be easily applied using the CONS DIHE command in CHARMM, with the sign in front of $U_{BP}(\chi)$ reversed. This will effectively cancel out the potential barrier between the rotamers of a given residue sidechain.

Before computing $\Delta G_{i\rightarrow j}^{\mathrm{site}}$ and $\Delta G_{i\rightarrow j}^{\mathrm{trans}}$, restraints were set up to restrict the translational freedom of the bound ion of interest. First, three points within the protein region were picked out. They were used along with the position of the bound ion to set up the restraints. We employed the same protocol as in the *Ligand Binder* module (*Jo et al., 2013*) in CHARMM-GUI to select the three protein anchor points, $p_1$, $p_2$, and $p_3$. The relative position of the bound ions to the protein can be defined

**Table 6.** Binding site residues and the fitted parameters $k_i$ and $\chi_{0,i}$.

| | $k_1$ | $\chi_{0,1}$ | $k_2$ | $\chi_{0,2}$ | $k_3$ | $\chi_{0,3}$ | $k_4$ | $\chi_{0,4}$ | $k_5$ | $\chi_{0,5}$ | $k_6$ | $\chi_{0,6}$ |
|---|---|---|---|---|---|---|---|---|---|---|---|---|
| M4 | | | | | | | | | | | | |
| E327p | -0.932 | 53.11 | 1.055 | 61.08 | 1.844 | 62.05 | -0.312 | 56.44 | 0.392 | 89.04 | 0.225 | 82.28 |
| M5 | | | | | | | | | | | | |
| Y771 | -2.19 | 62.51 | 1.219 | 89.27 | 2.46 | 60.91 | 0.662 | 40.85 | 0.721 | 44.75 | 0.639 | 50.42 |
| T774 | -4.25 | 35.82 | 2.437 | 101.3 | 3.602 | 61.71 | 1.128 | 44.86 | 1.071 | 91.96 | 0.857 | 76.09 |
| S775 | 3.068 | -26.37 | 0.124 | 38.67 | 1.291 | 62.22 | | | | | | |
| N776 | 2.664 | -166.23 | 0.918 | 87.15 | 1.709 | 62.85 | -0.53 | 71.79 | -0.495 | 50.76 | -0.406 | 75 |
| E779p | -2.508 | 29.17 | 1.792 | 88.01 | 2.591 | 62.28 | 0.487 | 24.81 | 0.476 | 73.74 | 0.462 | 53.74 |
| M6 | | | | | | | | | | | | |
| D804- | 3.895 | -172.95 | 0.489 | 80.04 | 1.897 | 60.56 | -0.197 | 82.91 | -0.271 | 82.85 | -0.193 | 79.47 |
| D804p | 2.968 | -157.98 | 0.532 | 81.76 | 1.835 | 60.45 | -0.247 | 31.05 | -0.296 | 89.84 | -0.278 | 83.37 |
| N804 | 2.981 | -160.18 | 0.371 | 84 | 1.451 | 59.42 | | | | | | |
| D808- | 3.215 | -159.94 | 0.21 | 57.87 | 1.744 | 59.13 | | | | | | |
| D808p | -2.298 | 55.97 | 1.204 | 66.88 | 2.44 | 62.53 | 0.746 | 37.74 | 0.715 | 55.74 | 0.627 | 63.72 |
| N808 | -2.344 | 29.85 | 1.771 | 63.6 | 2.654 | 64.99 | 0.914 | 39.19 | 0.73 | 58.37 | 0.63 | 61.7 |
| M8 | | | | | | | | | | | | |
| Q923 | -2.943 | 33.4 | 2.217 | 87.95 | 2.671 | 61.48 | 0.603 | 74.2 | 0.535 | 30.65 | 0.551 | 44.23 |
| D926- | 4.361 | -169.1 | 0.454 | 95.51 | 1.902 | 60.16 | -0.283 | 92.02 | -0.306 | 88.18 | -0.215 | 85.37 |
| D926p | -2.704 | 28.74 | 1.477 | 93.38 | 2.744 | 61.08 | 0.816 | 58.99 | 0.852 | 58.37 | 0.809 | 58.93 |
| N926 | 3.674 | -157.18 | 0.479 | 89.17 | 1.377 | 61.78 | -0.625 | 82.59 | | | | |

by the distance $r$ from $\mathbf{l_1}$ to $\mathbf{p_1}$, the angle $\theta$ between $\mathbf{l_1}$, $\mathbf{p_1}$ and $\mathbf{p_2}$, and the torsion $\psi$ along $\mathbf{l_1}$, $\mathbf{p_1}$, $\mathbf{p_2}$, and $\mathbf{p_3}$. The translational restraint potential took the form of

$$u_{trans} = \frac{1}{2}\left[k_r(r-r_0)^2 + k_\theta(\theta-\theta_0)^2 + k_\psi(\psi-\psi_0)^2\right]. \tag{4}$$

The force constant regarding distance (i.e., $k_r$) was set to 10 kcal/mol/Å$^2$, and the rest of the force constants were 200 kcal/mol/rad$^2$. The equilibrium values in $u_{trans}$ were taken from the 200 ps equilibration of the reduced binding site.

$\Delta G^{site}_{Na \to K}$ was computed with a set of FEP/H-REMD simulations. The translational restraint, $u_{trans}$, was applied to restrict the translational freedom of the ion. The Na$^+$ ion was changed alchemically to a K$^+$ with an alchemical coupling factor, $\lambda$. 16 $\lambda$ windows were used. Each $\lambda$ window included 8 replicas with the strength of boosting potential scaled from 0 to 1. Exchange attempts were made every 0.2 ps and were only allowed between neighboring replicas with different boosting potential strengths in the same $\lambda$ window and between the neighboring $\lambda$ windows with zero boosting potential. A total of 128 replica systems were included in the calculations. Each replica system was simulated for 2 ns. To compute $\Delta G^{trans}_{Na \to K}$ at a given binding site, two sets of simulations were set up, one with Na$^+$ at the binding site and the other with K$^+$ for the calculation of $\Delta G^{site}_{Na,\,trans}$ and $\Delta G^{site}_{K,\,trans}$. The translational and orientational restraints were decoupled gradually (*Beglov et al., 1994*) using a coupling factor $\lambda$. The $\lambda$ window and the boosting potential setup were similar to those used in the $\Delta G^{site}_{Na \to K}$ calculations. All the FEP/H-REMD calculations were performed using the REPDSTR module in CHARMM (*Jiang et al., 2010*). The output energies from the zero boosting potential systems were collected and processed using WHAM (*Kumar et al., 1992*). To compute the standard error of $\Delta\Delta G$, we divided the trajectories into 10 blocks. The standard deviations of the averaged $\Delta\Delta G$ from all the blocks are computed and reported in *Tables 4* and *5* and *Figures 3–5*. Cautions should be taken when comparing the calculated and experimentally measured $\Delta\Delta G$, because the latter usually

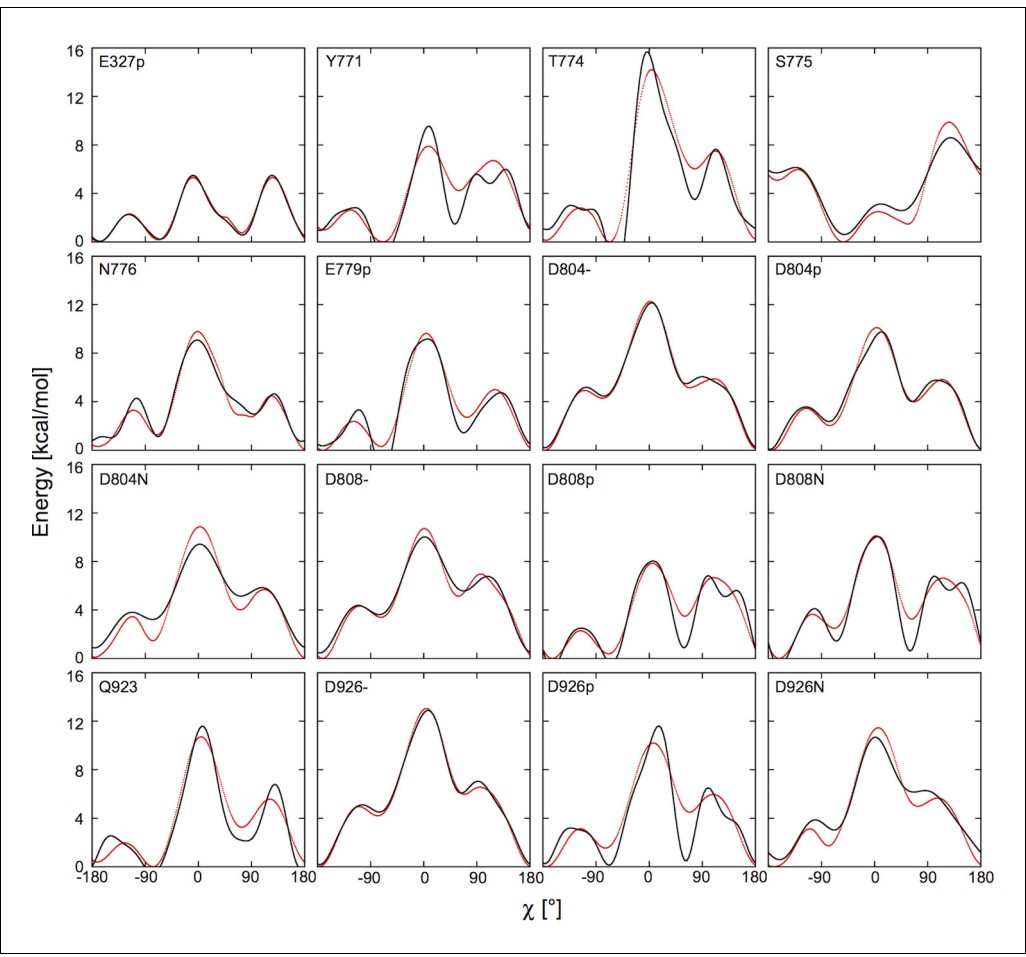

**Figure 7.** Fitting the potential of mean force $\mathcal{W}(\chi)$ (red) with the boosting potential, $U_{BP}(\chi)$ (black).

contains contributions from multiple states of the pump while the calculated $\Delta\Delta G$ is done using a single state.

## The bulk ion-water system and the calculation of $\Delta G_{Na\rightarrow K}^{bulk}$

The bulk system was generated by building a water sphere of 10 Å radius and centering it at the origin. The water sphere was made form pre-equilibrated water boxes with TIP3P water molecules. A $Na^+$ ion was placed at the center of the sphere. Any water molecules overlapping with the ion were deleted. The influence of the remaining bulk was approximated by the spherical solvent boundary potential (*Beglov et al., 1994*). The system was equilibrated for 200 ps at 303.15 K. Other simulation options were kept the same as described in the reduced binding site system. During the equilibration the position of the ion was restrained using a weak harmonic bias potential with a force constant of 0.5 kcal/mol/Å². Once equilibrated, the $Na^+$ was gradually changed to a $K^+$ using the PERT module in CHARMM with 11 λ windows. The simulation time for each λ window was 1 ns. The resulting $\Delta G_{Na\rightarrow K}^{bulk}$ is 18.34 kcal/mol after unbiasing the energy outputs with WHAM. The calculated $\Delta G_{Na\rightarrow K}$ for all the binding sites in the systems summarized in *Table 3* are shown in *Table 4*.

## D804 pKa calculations with explicit solvent

To further confirm the protonation state of D804, we evaluated its pKa shift with additional simulations in explicit solvent using the following formula:

$$\Delta pKa = \left(\Delta G_{site}^{deprot} - \Delta G_{bulk}^{deprot}\right)/2.303 k_B T. \tag{5}$$

$\Delta G_{\text{site}}^{\text{deprot}}$ and $\Delta G_{\text{site}}^{\text{deprot}}$ are the free energy change of aspartate deprotonation in the protein environment and in bulk water, respectively. The reduced system of E1_S1 (*Table 3*) was used to compute the deprotonation free energy of D804 at the ion binding site ($\Delta G_{\text{site}}^{\text{deprot}}$). To calculate the $\Delta G_{\text{bulk}}^{\text{deprot}}$, an aspartic acid residue with an acetylated N-terminus and an amidated C-terminus was put into a pre-equilibrated water sphere of 15 Å radius. As in the calculations of $\Delta G_{\text{Na} \rightarrow \text{K}}^{\text{bulk}}$ and $\Delta G_{\text{K} \rightarrow \text{Na}}^{\text{bulk}}$, the impact of the bulk solution beyond the current system was incorporated by the spherical solvent boundary potential (*Beglov et al., 1994*). Alchemical FEP calculations were carried out in both systems. The λ windows were evenly spaced to gradually deprotonate the aspartate. The numbers of windows used in the $\Delta G_{\text{site}}^{\text{deprot}}$ and $\Delta G_{\text{bulk}}^{\text{deprot}}$ calculations were 24 and 10 respectively. Each window was equilibrated for 200 ps and contained 5 ns of sampling. The calculated $\Delta G_{\text{site}}^{\text{deprot}}$ and $\Delta G_{\text{bulk}}^{\text{deprot}}$ are −44.8 kcal/mol and −51.9 kcal/mol, respectively. Using *Equation (5)*, the ΔpKa is 5.1, meaning the pKa of D804 is right shifted by 5.1 pK unit when it is in the Na$^+$/K$^+$ pump ion binding site. The final calculated pKa of D804 is 4.1 + 5.1 = 9.2 with 4.1 being the pKa value of an aspartate in bulk water (*Berg et al., 2002*), reinforcing the conclusion that D804 is protonated in the E$_1$ state of the pump.

## Transition pathway calculations

Three conformational transition pathways were generated based on the shared homology between the Na$^+$/K$^+$-pump and the Ca$^{2+}$ SERCA pump. One of such pathways connects states E$_2$(K$_2$) and P-E$_2$·K$_2$. The second connects states E$_2$(K$_2$) and K$_2$·E$_1$, and the third describes the transition between states Na$_3$·E$_1$-P and P-E$_2$·Na$_3$. First, A Cα-atom-only transition pathway for each of these was generated using the ANMPathway online server (http://anmpathway.lcrc.anl.gov/anmpathway.cgi) (*Das et al., 2014*) based on the SERCA pump crystal structures including 3B9B (*Olesen et al., 2007*), 1WPG (*Toyoshima et al., 2004*), and 1VFP (*Toyoshima et al., 2004*). Since both the Na$^+$/K$^+$-pump and the Ca$^{2+}$ SERCA pump are P-type ATPases, it is reasonable to assume that they have similar pumping mechanisms and they share the same set of states along the pump cycle. Although crystal structures of the Na$^+$/K$^+$-pump are scarce, multiple high-resolution structures of the SERCA pump in different states are available (*Karlsen et al., 2016*). Among them, 3B9B represents the outward facing P-E$_2$ state. All the other SERCA pump structures were aligned based on their transmembrane region Cα positions to the Na$^+$/K$^+$-pump and the Cα RMSD were computed to find those that resemble the structural states captured by the Na$^+$/K$^+$-pump crystal structures 3WGV (*Kanai et al., 2013*) and 2ZXE (*Shinoda et al., 2009*). The two SERCA pump structures representing states Na$_3$·E$_1$ATP (3WGV) and E$_2$(K$_2$) (2ZXE) are 1VFP and 1WPG, respectively. Each generated coarse-grained pathway is made of a sequence of structural snapshots containing Cα atoms only. These snapshots are called images and are distributed at equal RMSD intervals. The images from the coarse-grained pathways were then used in the all-atom targeted molecular dynamics (TMD) simulations to guide each transition. The starting system in the first and second pathway TMD simulations was taken from the well-equilibrated MD simulation system E2_S1 starting from the crystal structure 2ZXE. The third pathway was realized using the MD simulation system E1_S1 built from the crystal structure 3WGV, with the catalytic D369 phosphorylated. The protocol of the TMD simulations followed those published before (*Castillo et al., 2015*)

## Selectivity calculations along the pump cycle

Binding site ion selectivity was evaluated for nine systems representing different stages along the pump cycle. Among these systems are well-defined states: P-E$_2$·K$_2$, E$_2$(K$_2$), K$_2$·E$_1$, Na$_3$·E$_1$(ADP·Pi), and P-E$_2$·Na$_3$. The structures used for them are ① the generated P-E$_2$·K$_2$ model, ③ the equilibrated crystal structure 2ZXE, the equilibrated crystal structure 3WGV with ⑦ bound Na$^+$ and with ⑤ bound Na$^+$ replaced by K$^+$ at sites I and II, and ⑨ the generated P-E$_2$·Na$_3$ model. Several intermediate states were also included. Intermediate state ② is between states ① P-E$_2$·K$_2$ and ③ E$_2$(K$_2$), taken from the TMD simulations of the first path after image 35. State ④ was taken at image 33 along the transition from state ③ E$_2$(K$_2$) to ⑤ K$_2$·E$_1$. Along the same vein, state ⑧ represented the intermediate at image 29 between states ⑦ Na$_3$·E$_1$(ADP.Pi) and ⑨ P-E$_2$·Na$_3$. The intermediates were taken near the midpoints of the transitions. An additional state ⑥ K$_2$·E$_1$$^*$ with altered binding site residue protonation was also included to reflect the protonation state change upon the E$_2$ to E$_1$ transition. All

the states included in the selectivity calculations and their relative placement along the pump cycle can be seen in *Figure 5*.

To evaluate the selectivity at a given binding site in a system, three simulations were performed with slightly varied Lennard-Jones (LJ) parameters for the ion of interest. In the first simulation, the parameters of the ion were unaltered. A linear interpolation of the LJ and the NBFIX parameters between $Na^+$ and $K^+$ was made. In the second simulation, the LJ and NBFIX parameters of the ion were replaced by those of a Na/K hybrid at the middle point of the interpolation. In the last simulation, the ion's parameters were completely changed to represent the other ion type, either $K^+$ or $Na^+$, depending on the starting ion type. 10 ns trajectories were generated for each simulation using the simulation package OpenMM6.2 (*Eastman et al., 2013*). Energies were calculated from each trajectory using two parameter sets. The difference was fed into the WHAM equation (*Kumar et al., 1992*) and the free energy changes upon mutating the ion to the intermediate as well as upon mutating the intermediate to the other ion type were solved self-consistently. The sum of the two free energies gives the $\Delta G^{site}$. The same strategy was used to compute the $\Delta G^{bulk}$ in water ($\Delta G^{bulk}$ kcal/mol). The binding free energy difference is given by $\Delta\Delta G_{bind} = \Delta G^{site} - G^{bulk}$. The selectivity of binding site with different occupancies was also included in the calculations. The $\Delta\Delta G_{bind}$ are shown in *Table 5*.

## NAMD simulation protocol

The initial relaxation and the restraint-free equilibration of the membrane pump systems were performed using the NAMD2.9 (*Phillips et al., 2005*) simulation package with the input scripts from the CHARMM-GUI *membrane builder* module. The NAMD simulation temperature was set to 303.15 K using Langevin Dynamics with a damping coefficient of 10 $ps^{-1}$ during the relaxation and 1 $ps^{-1}$ during the restraint-free equilibration. The van der Waals interactions were smoothly switched off at 10–12 Å by a force switching function (*Steinbach et al., 1994*) and the electrostatic interactions were calculated using the particle-mesh Ewald method with a mesh size of ~1 Å.

## Anton simulation protocol

After a short equilibration with NAMD, each all-atom system listed in *Table 3* were subjected to a hundred ns scale long production without any restraints using the special-purpose supercomputer Anton (*Shaw et al., 2009*). The volume of the periodic cell was kept constant and the temperature was set to 303.15K using the Nosé-Hoover thermostats (*Martyna et al., 1992*). The lengths of all bonds involving hydrogen atoms were constrained using M-SHAKE (*Kräutler et al., 2001*). The cutoff of the van der Waals and short-range electrostatic interactions was set to an optimal value suggested by the Anton guesser script, guess_chem. Long-range electrostatic interactions were evaluated using the k-space Gaussian split Ewald method (*Shan et al., 2005*) with a 64 × 64 × 64 mesh. The integration time step was 2 fs. The r-RESPA integration method (*Tuckerman et al., 1992*) was employed and long-range electrostatics were evaluated every 6 fs.

## OpenMM simulation protocol

Simulations used to compute ion selectivity along the pump cycle were conducted using the simulation package OpenMM6.2 (*Eastman et al., 2013*). The constant pressure and temperature (NPT) ensemble was used for these simulations. The pressure was maintained at 1 atm with a Monte Carlo barostat, which attempts to adjust the system volume every 0.2 ps. The Langevin dynamics algorithm with a 1.0 $ps^{-1}$ friction coefficient was used to hold the simulation temperature, which is at 303.15 K. The lengths of all bonds involving hydrogen were fixed and the integration time step was set to 2 fs. A force switching function was applied from 10 to 12 Å to gradually turn off the van der Waals interactions and the particle-mesh Ewald method with an error tolerance of 0.0005 was used to evaluate the electrostatic interactions.

## Simulation force field parameters

The same force field parameters were used in all other simulations in this study. The PARAM27 all-atom force field of CHARMM (*MacKerell et al., 1998*) with a modified version of dihedral cross-term correction (*Mackerell et al., 2004*) was used for the protein and the C36 lipid force field

(*Klauda et al., 2010*) was used for POPC. Water molecules were modeled with the TIP3P potential (*Jorgensen et al., 1983*).

## Acknowledgements

This work was supported by grant U54-GM087519 (to BR) and R15(NS081570-01A to PA) from NIH, and by grant MCB-1515434 (to PA and BR) from NSF. Anton computer time was provided by the National Center for Multiscale Modeling of Biological Systems (MMBioS) through Grant P41GM103712-S1 from the National Institutes of Health and the Pittsburgh Supercomputing Center (PSC). The Anton machine at PSC was generously made available by D.E Shaw Research.

## Additional information

### Funding

| Funder | Grant reference number | Author |
| --- | --- | --- |
| National Institutes of Health | P41GM103712-S1 | Huan Rui<br>Benoît Roux |
| National Science Foundation | MCB-1515434 | Pablo Artigas<br>Benoît Roux |
| National Institutes of Health | R15, NS081570-01A | Pablo Artigas |
| National Institutes of Health | U54-GM087519 | Benoît Roux |

The funders had no role in study design, data collection and interpretation, or the decision to submit the work for publication.

### Author contributions

HR, Conception and design, Acquisition of data, Analysis and interpretation of data, Drafting or revising the article; PA, Analysis and interpretation of data, Drafting or revising the article; BR, Conception and design, Drafting or revising the article

### Author ORCIDs

Huan Rui, http://orcid.org/0000-0002-6459-9871
Benoît Roux, http://orcid.org/0000-0002-5254-2712

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
