## [Decision Letter]

Thank you for submitting your article "Binding site protonation and self-correcting occlusion control the Na+/K^+^-pump selectivity" for consideration by *eLife*. Your article has been reviewed by three peer reviewers, including Erik Lindahl (Reviewer #3) and Nir Ben-Tal (Reviewer #1), who is a member of our Board of Reviewing Editors, and the evaluation has been overseen by Richard Aldrich as the Senior Editor.

The reviewers have discussed the reviews with one another and the Reviewing Editor has drafted this decision to help you prepare a revised submission.

Summary:

The study examines the physicochemical determinants of binding site selectivity in the Na+/K^+^-ATPase using molecular dynamics simulations. Various conformation of the ATPase, corresponding to different states along the pump cycle are examined, including the crystal structures of its two main conformations, as well as several models of intermediate states based on the Ca^2+^ SERCA pump, and models derived from interpolating between well-defined states. Various protonation states of titratable residues in the pump's binding sites are energetically examined using molecular dynamic simulations as different resolutions. The results suggest that the protonation states of three aspartate residues in the binding sites (D804, D808 and D926) alter upon switching between conformations. The results indicate that the different selectivity of the binding sites emerges from the change in the protonation states. Using reduced structural models of the binding sites, this hypothesis is supported by calculating the difference in free energy between the pump bound to Na^+^ vs. K^+^ ions along the transport cycle. The authors conclude that initially the binding sites are only weakly selective for either Na^+^ or K^+^, and selectivity emerges as collective effect due to the (indirect) interaction of new ions with the ones that are already bound (a knock-on effect); the pump shifts towards K^+^ selectivity in the outward open state, and towards Na^+^ selectivity in the inward open state. Binding of the wrong type of ion increases the free energy barrier towards occlusion, preventing the transport cycle to proceed, thereby preventing the pump from operating in the opposite (wrong) direction.

The manuscript addresses an interesting and important question. It proposes a simple mechanism that dictates the shift in the selective properties of the Na+/K^+^-pump binding sites. However, based on the current draft of the manuscript it is difficult to decide whether the work is novel and solid enough to justify publication in *eLife*. The summary reflects a long discussion among the reviewers in an effort to resolve this. The author should submit a revised draft only if they are certain that they can address all the issues raised.

Essential revisions:

1) It is difficult to figure out which structures the various steps in Figure 5 are based on. Perhaps:

State 1 and 9 correspond to 3B9B (open SERCA);

State 2 calculated transition to;

State 3 corresponds to 1WPG (occluded SERCA) – but why not 2ZXE (occluded NKA)?

State 4 calculated transition to;

State 5, unclear: called K_2_E_1_. Maybe 1VFP with different protonation pattern and K instead of Na?

State 6 corresponds to 1VFP (Ca occluded SERCA) – but with K rather than Na?

State 7 corresponds to 3WGV (occluded NKA);

State 8 calculated transition to 9;

The authors should make perfectly clear which structures are used for which states.

2) The pump cycle should be based on all the relevant structures. Structures have been published of SERCA in E_2_ (3W5C) and the open ready-to-bind Ca (4H1W) in 2013. These important steps are missing here, or guesses are attempted based on the Ca-occluded 1VFP structure, but that did not predict the structure for H and Ca exchange. The structures are not mentioned. (A nice and recent overview of P-type structures can be found here http://www.ncbi.nlm.nih.gov/pubmed/26695058).

Also, it would be preferable to use 4HYT (ouabain-bound NKA, mentioned as recent but it is from 2013) rather than 3B9B. At least the authors could show an overlay of the ion binding sites from 4HYT and their model.

3) Standard errors estimates should be added to the energy values, and conclusions that have been based on energy values with too large 'error bars' should be reconsidered.

4) If the calculations correctly describe the binding free energy of the pump cycle, they would also need to explain the ion release on the opposite side. Can they show this?

5) The simulations and free energy calculations assume a starting point in which all the pump's binding sites are occupied by a certain set of ions. The authors explicitly claim that selectivity emerges from the increased free energy barrier associate with binding of the wrong ion to a site. However, from a biological point of view to argue that selectivity is governed solely by such mechanism is puzzling, for it is inefficient. The mechanism proposed here beautifully explains the different selective properties of the pump's binding sites in its two conformations, and it is possibly a "last resort mechanism", designed, as the authors claim, to prevent the transport of the wrong kind of ion if it reaches the binding site. However one would expect that for the most part, under physiological conditions, the pump will prevent the wrong ions from reaching the binding site to begin with. The anticipation is that a fast machine, such as this pump, should follow a particularly efficient mechanism, rather than the one proposed here, which figures out that there is issue with selectivity only after binding of several ions of the wrong type. The manuscript does not account for the ions transition into the binding sites and other possible energetic barriers they might encounter on the way. This issue should, at the very least, be discussed.

6) In view of item 5, the authors should revise the title to reflect the fact that selectivity is presumably determined primarily by other mechanisms, and the suggested mechanism here is only a 'safe guard'.

---

## [Author Response]

*Essential revisions:*

*1) It is difficult to figure out which structures the various steps in Figure 5 are based on. Perhaps: State 1 and 9 correspond to 3B9B (open SERCA); State 2 calculated transition to; State 3 corresponds to 1WPG (occluded SERCA) – but why not 2ZXE (occluded NKA)? State 4 calculated transition to; State 5, unclear: called K2E1. Maybe 1VFP with different protonation pattern and K instead of Na? State 6 corresponds to 1VFP (Ca occluded SERCA) – but with K rather than Na? State 7 corresponds to 3WGV (occluded NKA); State 8 calculated transition to 9; The authors should make perfectly clear which structures are used for which states.*

We modified the “*Selectivity calculations along the pump cycle” section in* Materials and methods to introduce the structures used for the calculations of ΔΔ*G* along the pump cycle. The updated text reads as follows:

“The structures used for them are: 1) the generated P-E_2_.K_2_ model, 3) the equilibrated crystal structure 2ZXE, the equilibrated crystal structure 3WGV with 7) bound Na^+^ and with 5) bound Na^+^ replaced by K^+^ at sites I and II, and 9) the generated P-E_2_.Na_3_ model. […] All the states included in the selectivity calculations and their relative placement along the pump cycle can be seen in Figure 5.”

*2) The pump cycle should be based on all the relevant structures. Structures have been published of SERCA in E_2_ (3W5C) and the open ready-to-bind Ca (4H1W) in 2013. These important steps are missing here, or guesses are attempted based on the Ca-occluded 1VFP structure, but that did not predict the structure for H and Ca exchange. The structures are not mentioned. (A nice and recent overview of P-type structures can be found here http://www.ncbi.nlm.nih.gov/pubmed/26695058)*

*Also, it would be preferable to use 4HYT (ouabain-bound NKA, mentioned as recent but it is from 2013) rather than 3B9B. At least the authors could show an overlay of the ion binding sites from 4HYT and their model.*

First of all, we apologize for the confusion. Detailed descriptions on how the transition pathways are generated based on the Ca^2+^ SERCA pump structures are not included. We felt that it was not necessary because the complete protocol was described in our previous publication (Castillo et al., 2015). We hope that adding the sentences below in the manuscript could clarify some of the reviewers’ concerns:

“Since both the Na^+^/K^+^-pump and the Ca^2+^ SERCA pump are P-type ATPases, it is reasonable to assume that they have similar pumping mechanisms and they share the same set of states along the pump cycle. […] The two SERCA pump structures representing states Na_3_.E_1_ATP (3WGV) and E_2_(K_2_) (2ZXE) are 1VFP and 1WPG, respectively.”

We are aware of the 4HYT structure, but did not use it in the free energy calculations because it is in a P-E_2_-like state with ouabain bound blocking the extracellular access rather than a true outward open state. The generated P-E_2_ models and the crystal structure 4HYT show similarity, especially in the transmembrane region, where the backbone RMSD is 2.7 Å (see the Extended Data Figure 5 in our previous publication on the pump (Castillo et al., 2015)). This is mentioned in the main text of the manuscript in the Result section titled “*Ion selectivity along the pump cycle*”.

“The models appear sensible as they are structurally comparable to the recently published P-E_2_ like structures of the Na^+^/K^+^-pump (Gregersen et al., 2016; Laursen et al., 2013) and the vanadate-Inhibited, P-E_2_ mimic of the Ca^2+^ SERCA pump (Clausen et al., 2016).”

The K^+^-loaded P-E_2_ model also reproduces the experimentally measured gating charge. Therefore, we think the current P-E_2_ models are reasonable and can be used compute the free energy difference of Na^+^ and K^+^ binding.

*3) Standard errors estimates should be added to the energy values, and conclusions that have been based on energy values with too large 'error bars' should be reconsidered (see comment 1 by Reviewer 3).*

We updated the relevant figures and tables.

*4) If the calculations correctly describe the binding free energy of the pump cycle, they would also need to explain the ion release on the opposite side. Can they show this?*

Yes. The free energy values can be used to explain cytoplasmic release of the K^+^ as well. Once the pump opens to the cytoplasmic side, protonation states of the binding site aspartates change and the binding sites go from highly K^+^ selective in the occluded state to relatively non-discriminating. Even though rebinding of the two K^+^ is possible given the much higher K^+^ concentration in the cytoplasm, the rate of the backward transition to the occluded state is slow. At the same time, the binding of K^+^ to the site III is unlikely. Because occupancy of this site is the prerequisite of phosphorylation and occlusion of E_1_, the pump cycle cannot go forward. All the factors favor the cytoplasmic release of K^+^ and binding of Na^+^, which keep pump cycle forward. To emphasize the point, we add the following sentences in the main text:

“When the pump is in this state, site I is Na^+^ selective and site II is only weakly K^+^ selective. Together, the E_2_ to E_1_ structural transition and the protonation state change promote the release of K^+^ to the cytoplasmic side.”

*5) The simulations and free energy calculations assume a starting point in which all the pump's binding sites are occupied by a certain set of ions. The authors explicitly claim that selectivity emerges from the increased free energy barrier associate with binding of the wrong ion to a site. However, from a biological point of view to argue that selectivity is governed solely by such mechanism is puzzling, for it is inefficient. The mechanism proposed here beautifully explains the different selective properties of the pump's binding sites in its two conformations, and it is possibly a "last resort mechanism", designed, as the authors claim, to prevent the transport of the wrong kind of ion if it reaches the binding site. However one would expect that for the most part, under physiological conditions, the pump will prevent the wrong ions from reaching the binding site to begin with. The anticipation is that a fast machine, such as this pump, should follow a particularly efficient mechanism, rather than the one proposed here, which figures out that there is issue with selectivity only after binding of several ions of the wrong type. The manuscript does not account for the ions transition into the binding sites and other possible energetic barriers they might encounter on the way. This issue should, at the very least, be discussed.*

First, we need to clarify that the selectivity of the pump is not solely governed by the “self-correcting” mechanism. The occluded states are indeed highly selective while the states that are open to the extracellular and cytoplasmic sides are less so. The “self-correcting” mechanism is one way for the pump to achieve this selectivity difference between the states.

The pump is prone to error when it tries to bind ions from the extracellular and cytoplasmic solutions. Take the extracellular ion binding for example – in order for K^+^ to have an advantage at binding in the presence of a Na^+^ concentration that is *N* times higher ([K+]o[Na+]o=1N), there needs to be at least *N* fold difference in *K*_D_ to guarantee a 1:1 ratio between [ENa] and [EK]:KD,KKD,Na=[K+]o[E][EK][Na+]o[E][ENa]=[K+]o[Na+]o⋅[ENa][EK]

If the pump were to only pick out K^+^ from the extracellular media, the K_D,K_ binding should be much smaller than K_D,Na_. Experimentally measured K_D,K_ at the extracellular side is on the order of 1 mM in mammalian cell Na^+^/K^+^-pump in the presence of Na^+^, while that of K_D,Na_ is on the order of 100 mM (Nakao and Gadsby, 1989; De Weer et al. 2001). Given that the extracellular [Na^+^] is about 25 times that of [K^+^], for every 5 ion-loaded pumps, 1 would have the wrong configuration at the binding pocket. The “self-correcting” mechanism is a strategy evolved in the pump to cope with the problem that comes with the not-quite-specific external and cytoplasmic binding.

From an evolution point of view, it is not surprising that the outward and inward open states of the pump are less selective than the occluded states. When the pump is in an inward or outward open state, the binding pocket is more voluminous and hydrated than it is in the occluded state. Because of the spatial restriction in the occluded state-binding site, the bound ions would feel the impact from mutations more strongly than when they are bound in a larger pocket. During the evolution process, it would be easier to create a tightly packed, ion specific pocket than to make a large aqueous compartment selective towards ions of similar size and charge.

The Na^+^/K^+^-pump is not a fast molecular machine. The estimated turnover rate is less than 100 per second and decreases even further as the transmembrane voltage becomes more negative. This is perhaps because the pump spends time wait for the correct ion configuration to be achieved at the binding site, even though the idea of a self-correcting ion selectivity mechanism seems counterintuitive and it is more natural to think of a pump that follows a particularly efficient mechanism where it prevents the wrong ions from reaching the binding site under physiological conditions. The truth could be that the pump has not been evolutionarily optimized to be a fast machine, but an energetically efficient one.

We agree with the reviewers that other possible energy barriers encountered by the ions when they move in and out of the binding should be discussed. Although not shown in the manuscript, we observed ions, both K^+^ and Na^+^, can access the binding pocket within tens of ns in MD simulations of the pump in its inward and outward open states. As a result, this is not likely to contribute to the observed binding affinity difference substantially. This is added to the updated Discussion session:

“Other energetic barriers, like pathways the ions use to travel to the binding site, could contribute to the ion binding specificity, it is not likely the case here as both Na^+^ and K^+^ can access the binding pocket in the simulations of the E_1_ and P-E_2_ states without ions bound.”

*6) In view of item 5, the authors should revise the title to reflect the fact that selectivity is presumably determined primarily by other mechanisms, and the suggested mechanism here is only a 'safe guard'.*

As mentioned above, we do think that the “self-correcting” mechanism is an essential element of the primary mechanism the pump uses to transport ions. To highlight this point, we changed the title to “The selectivity of the Na+/K^+^-pump is controlled by binding site protonation and self-correcting occlusion”.